# PRED: Pre-training via Semantic Rendering on LiDAR Point Clouds

**Hao Yang**[1,3]   **Haiyang Wang**[1]   **Di Dai**[2]   **Liwei Wang**[1,2]
[1]Center for Data Science, Peking University
[2]National Key Laboratory of General Artificial Intelligence, School of Intelligence
Science and Technology, Peking University   [3]Pazhou Lab
{haoy@stu, wanghaiyang@stu, didai@stu, wanglw@cis}.pku.edu.cn

## Abstract

Pre-training is crucial in 3D-related fields such as autonomous driving where point cloud annotation is costly and challenging. Many recent studies on point cloud pre-training, however, have overlooked the issue of incompleteness, where only a fraction of the points are captured by LiDAR, leading to ambiguity during the training phase. On the other hand, images offer more comprehensive information and richer semantics that can bolster point cloud encoders in addressing the incompleteness issue inherent in point clouds. Yet, incorporating images into point cloud pre-training presents its own challenges due to occlusions, potentially causing misalignments between points and pixels. In this work, we propose PRED, a novel image-assisted pre-training framework for outdoor point clouds in an occlusion-aware manner. The main ingredient of our framework is a Birds-Eye-View (BEV) feature map conditioned semantic rendering, leveraging the semantics of images for supervision through neural rendering. We further enhance our model's performance by incorporating point-wise masking with a high mask ratio (95%). Extensive experiments demonstrate PRED's superiority over prior point cloud pre-training methods, providing significant improvements on various large-scale datasets for 3D perception tasks. Codes will be available at https://github.com/PRED4pc/PRED.

## 1   Introduction

Pre-training is a fundamental task in deep learning, serving as a powerful tool to harness the potential of copious unlabeled data and enhance the results of downstream tasks. This is especially critical in 3D-related fields, including autonomous driving, where the annotation of point clouds is both laborious and expensive, contrasting with the relative ease of amassing large volumes of unlabeled point cloud data (59). To tackle this challenge, we present **PRED** (**PRE**-training via semantic ren**D**ering), a novel framework designed to pre-train point cloud processors using multi-view images.

With the tremendous advancements in masked signal modeling within the realms of image and natural language processing (16; 12; 2), recent studies (55; 30; 62; 17; 61; 6) have delved into applying masked auto-encoders to pre-training in the field of point clouds. These works firstly apply patch masking to the input point clouds, then employ an encoder-decoder framework to reconstruct the point clouds, as depicted in Figure 1(a). Apart from masked auto-encoders, other researchers (52; 59; 64; 22; 34; 8) have explored point cloud pre-training through contrastive learning. Here, positive pairs are generated by applying varied data augmentations to the point clouds, as demonstrated in Figure 1(b). Despite these significant strides, there remains a blind spot regarding the inherent incompleteness of point clouds, which is a ubiquitous issue in outdoor LiDAR datasets. For example in nuScenes (4), a large-scale outdoor LiDAR dataset, over 30% of the labeled objects contain fewer

37th Conference on Neural Information Processing Systems (NeurIPS 2023).

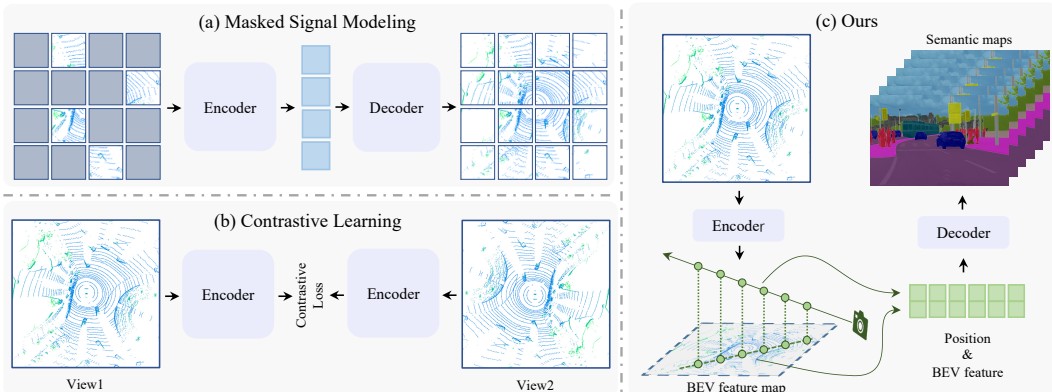

Figure 1: **Comparison of our framework with previous point cloud pre-train methods.** Existing pre-train works can be broadly categorized into (a) mask signal modeling and (b) contrastive learning. (c) We propose a novel framework that pre-trains point cloud encoder through semantic rendering.

than five points. As illustrated in Figure 2(a), this incompleteness introduces ambiguity into point cloud reconstruction, thereby potentially impacting the quality of the training process.

Images, compared to point clouds, embody more comprehensive information and enriched semantics. Several studies (6; 8; 34) have sought to offset the limitations of outdoor point clouds by incorporating images into point cloud pre-training. However, these works typically align point clouds with images through a straightforward point-to-pixel projection, neglecting to account for occlusion where objects in point clouds may not be visible in the image due to misalignment between LiDAR and camera. This oversight can lead to mismatches between points and pixels, as depicted in Figure 2(b), potentially hindering the effectiveness of pre-training. Hence the principal challenge we address in this paper is an occlusion-aware, image-assisted pre-training framework for outdoor point clouds.

In this study, we introduce a novel pre-training framework, PRED, designed to tackle the problem of reconstruction ambiguity and occlusion by integrating image information through semantic rendering, as illustrated in Figure 1(c). Neural rendering has demonstrated substantial success in learning 3D implicit representations from 2D images within the sphere of 3D generation (28; 31; 40). This inspires us to learn point cloud representations from image semantics via neural rendering. To accomplish this, we first extract BEV feature map from the point cloud using an encoder. Then, we render the semantic maps from image views, conditioned on the BEV feature map. The entire process is supervised by image semantics, thereby circumventing the introduction of reconstruction ambiguity and enabling effective occlusion management by assigning a reduced weight to occluded points during volume rendering. Moreover, we find the critical role of point-wise masking, employing a substantially higher mask ratio of 95% compared to the 75% utilized in previous patch-wise masking methods (16; 55).

We conduct extensive experiments on several outdoor LiDAR datasets, applying a variety of baselines and encoders to appraise the effectiveness of our pre-training method. The experimental results consistently showcase the superiority of our approach, outperforming both training from scratch and recent concurrent point cloud pre-training methods. Our contributions can be summarized as follows: 1) We introduce PRED, a novel pre-training framework for outdoor point clouds via semantic rendering. 2) We incorporate point-wise masking with a high mask ratio to enhance the performance of PRED. 3) Our approach significantly bolsters the performance of various baselines on large-scale nuScenes (4) and ONCE (26) datasets across various 3D perception tasks.

## 2 Related Work

**3D Perception on Point Clouds.** 3D perception methods generally fall into two categories: point-based and voxel-based. Point-based methods (9; 24; 32; 38; 48; 57; 58) directly extract features and

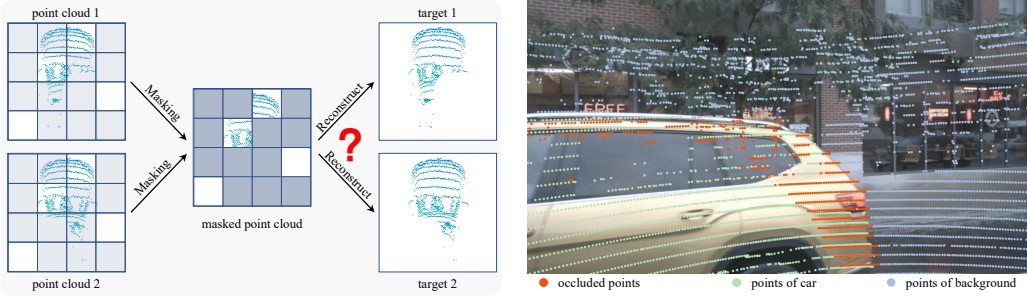

(a) **Reconstruction ambiguity.**     (b) **Mismatch between points and pixels.**

Figure 2: **(a)** The two point clouds on the left side of 2(a) display varied levels of incompleteness. Nevertheless, after masking, these two point clouds become strikingly similar, resulting in a single input corresponding to dual targets during reconstruction, thereby instigating ambiguity in training. **(b)** We present the projection of the LiDAR point cloud from the image view. The red dots belong to the background. However, due to occlusion, these points are mistakenly projected onto the car.

predict 3D objects from raw point clouds, while voxel-based methods (11; 13; 15; 36; 37; 39; 45; 54; 60) streamline the process by discretizing these points into regular voxels or pillars for processing with 3D or 2D convolutional networks. Recently, integrating the Transformer (43) into point cloud processing has yielded notable results (27; 14; 42; 47; 49). In this work, we show that a wide range of modern 3D perception methods can benefit from our pre-training framework.

**Pre-Training for Point Clouds.** Mask-based (61; 30; 62; 46) and contrast-based (22; 52; 64; 23; 33) methods are currently the leading paradigms for point cloud pre-training. However, these techniques often neglect the inherent incompleteness of outdoor point clouds. To address this, some studies (6; 8; 34) try to integrate images for a more comprehensive perspective. Though impressive, they overlook potential occlusion when aligning points with images. To counter these challenges, we introduce PRED, an effective pre-training paradigm that capitalizes on images through semantic rendering.

**Neural Rendering.** The implicit representations encoded by a neural network (28; 50; 29; 3; 41; 5; 56; 51) have gained a lot of attention recently, where the geometry and appearance in 3D space are encoded by implicit neural representation supervised from 2D images. NeuS (50) constrains the scene space as a signed distance function and applies volume rendering to train this representation. In this work, we apply a NeuS-like rendering method, but different from learning an implicit representation of a single scene, we aim to learn generalized point cloud representations for pre-training.

**Volumetric Rendering for Perception Tasks.** Recently, some works (53; 44) have attempted to introduce volume rendering into perception tasks. Ponder (19) shares a similar spirit to our work. However, Ponder focuses on indoor scenes, where point clouds often contain color information, facilitating color-based pre-training supervision. In contrast, our work addresses outdoor environmentsspecifically, autonomous drivingwhere point clouds are typically LiDAR-derived and colorless. Consequently, Ponder is not applicable due to the absence of color data. In this context, we propose semantic rendering. Unlike Ponder's color-based rendering, our approach capitalizes on the semantic consistency between point clouds and images, offering a distinct strategy for point cloud pre-training.

## 3 Methodology

We introduce **PRED**, a novel image-assisted pre-training framework designed for outdoor point clouds, taking into account occlusion. The comprehensive framework is depicted in Figure 3, with subsequent sections delving into the specifics of our method.

### 3.1 Revisiting NeuS

NeuS (50) is a neural rendering technique employed for single-scene 3D reconstruction. In this approach, the 3D scene is represented by two functions: a signed distance function (SDF) $d : \mathbb{R}^3 \rightarrow$

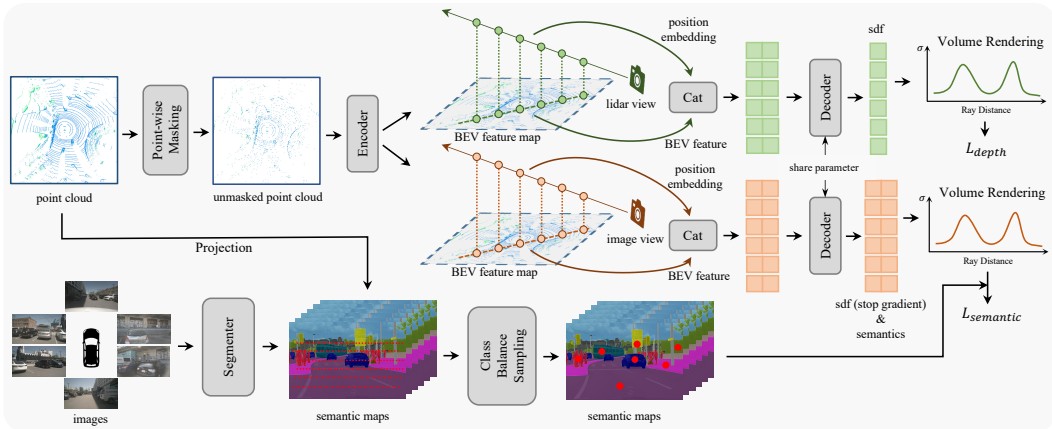

Figure 3: **Pipeline of our PRED.** Firstly, we apply point-wise masking to the input point cloud, subsequently feeding the remaining points into the encoder to generate the BEV feature map. Semantic rendering is then performed on the BEV feature map, supervised by both semantic and depth loss. For the semantic loss, we compute the cross-entropy loss between the rendered semantics and the pseudo labels, which are predicted by an image segmenter. During pre-training, batches are sampled from the point cloud projections on the image plane, utilizing class balance sampling. Upon the completion of pre-training, we employ the encoder for downstream tasks.

$\mathbb{R}$, which maps each spatial position $\mathbf{r} \in \mathbb{R}^3$ to its distance from the object's surface, and a color function $c : \mathbb{R}^3 \times \mathbb{S}^2 \to \mathbb{R}^3$, encoding the color of each point in the scene based on a specific viewing direction $\mathbf{v} \in \mathbb{S}^2$. Given a pixel with a ray emitted from it denoted as $\{\mathbf{r}(t) = \mathbf{o} + t\mathbf{v} \mid t \geq 0\}$, where $\mathbf{o}$ is the camera center and $\mathbf{v}$ is the unit direction vector of the ray, NeuS computes the color for this pixel by integrating the color values along the ray using the following equation:

$$C(\mathbf{o}, \mathbf{v}) = \int_0^{+\infty} w(t)c(\mathbf{r}(t), \mathbf{v})\mathrm{d}t, \tag{1}$$

where $C(\mathbf{o}, \mathbf{v})$ is the pixel's output color, $w(t)$ represents a weight function for the point $\mathbf{r}(t)$. To derive an unbiased weight function $w(t)$, NeuS computes the weight function as follows:

$$w(t) = \exp\left(-\int_0^t \rho(u)\mathrm{d}u\right)\rho(t), \quad \rho(t) = \max\left(\frac{-\frac{\mathrm{d}\Phi_s}{\mathrm{d}t}(d(\mathbf{r}(t)))}{\Phi_s(d(\mathbf{r}(t)))}, 0\right), \tag{2}$$

where $\Phi_s(x)$ is the sigmoid function $\Phi_s(x) = (1 + e^{-sx})^{-1}$, and $s$ is a trainable parameter. NeuS is then trained by minimizing the discrepancy between rendered prediction and ground truth colors.

## 3.2 Pre-Training via Semantic Rendering

We apply Equations 1 and 2 as our rendering method. A simple approach would be to use the image's RGB values as supervision for neural rendering like NeuS (50), but it failed to converge due to the point cloud's color absence. Consequently, we opt for semantic supervision. Unlike the lack of color information, the point cloud contains semantic details, which can be further enhanced through the image, resulting in better convergence during training. As demonstrated in Figure 3, our proposed method employs an encoder to transform input point clouds into a BEV feature map, and a decoder to map spatial positions in the 3D space, represented as $\mathbf{r} \in \mathbb{R}^3$, to their corresponding semantic predictions and signed distances to objects, conditioned on the BEV feature map.

Specifically, let $\mathcal{P} \in \mathbb{R}^{N_p \times 3}$ and $\mathcal{I} \in \mathbb{R}^{N_i \times H_i \times W_i \times 3}$ represent the input point clouds with $N_p$ points and corresponding $N_i$ images with width $W_i$ and height $H_i$, respectively. Firstly, we encode the point clouds $\mathcal{P}$ into a BEV feature map $\mathbf{F} \in \mathbb{R}^{H \times W \times C}$ with the encoder $E : \mathcal{P} \to \mathbf{F}$, where $H$, $W$, and $C$ signify the height, width, and number of channels of the feature map, respectively. Subsequently, we sample a batch of pixels from the images $\mathcal{I}$ and carry out semantic rendering for pre-training.

**Semantic Rendering.** Here, we use the pixel $\mathbf{u} = [u, v]$ as an example to illustrate the process. We denote the camera intrinsic and extrinsic parameters respectively by $\mathbf{K} \in \mathbb{R}^{3 \times 3}$ and $\mathbf{E} = [\mathbf{R}, \mathbf{t}] \in \mathbb{R}^{3 \times 4}$. A ray emitting from the pixel $\mathbf{u}$ can be characterized as a line $\mathcal{R} = \mathbf{t} + \delta \, \mathbf{R} \mathbf{K}^{-1} [u, v, 1]^\top$ in world coordinates parametrized by $\delta$. We then sample a sequence of points $\mathbf{r}_i = \mathcal{R}(\delta_i)$ along the ray $\mathcal{R}$. For each point, we utilize a decoder $D$ to predict its semantics and signed distance, given by

$$(\mathbf{s}(\mathbf{r}), d(\mathbf{r})) = D(\mathbf{Concat}(\mathbf{F}(\mathbf{r}), \mathbf{PE}(\mathbf{r}))), \tag{3}$$

where $\mathbf{s}(\mathbf{r}) \in \mathbb{R}^{N_c}$ and $d(\mathbf{r})$ denote the semantic prediction with $N_c$ classes and signed distance of point $\mathbf{r}$, $\mathbf{F}(\mathbf{r})$ symbolizes the feature extracted from the BEV feature map where $\mathbf{r}$ is projected, and $\mathbf{PE}(\mathbf{r})$ is the cosine position embedding of $\mathbf{r}$. Finally, we aggregate the semantic predictions along the ray $\mathcal{R}$ employing Equations 1 and 2, but in a discrete format:

$$\hat{\mathbf{s}} = \sum_{i=1}^{N_s} \prod_{j=1}^{i-1} (1 - \alpha_j) \, \alpha_i \mathbf{s}(\mathbf{r}_i), \quad \alpha_i = \max \left( \frac{\Phi_s\left(d(\mathbf{r}_i)\right) - \Phi_s\left(d(\mathbf{r}_{i+1})\right)}{\Phi_s\left(d(\mathbf{r}_i)\right)}, 0 \right), \tag{4}$$

where $N_s$ represents the number of sampled points. Throughout this process, occluded points will be allocated a low weight (50), thus mitigating the adverse effect of occlusion.

**Semantic supervision.** Given the absence of ground-truth semantic labels for images, we resort to DeepLabv3 (7) trained on CityScape (10) for predicting pseudo semantic labels for pixel $\mathbf{u}$, denoted as $\bar{\mathbf{s}} \in \mathbb{R}^{N_c}$. Intuitively, the accuracy of $\bar{\mathbf{s}}$ is linked to its maximum value: larger maximum values indicate more confident predictions with fewer errors. Hence, to offset the impact of inaccurate labels, this maximum value is incorporated as a weight in the loss calculation. Ultimately, our loss function adopts the form of cross-entropy loss:

$$\mathcal{L}_{semantic} = \frac{1}{|\mathcal{U}|} \sum_{\mathbf{u} \in \mathcal{U}} - \max(\bar{\mathbf{s}}) \sum_{i=1}^{N_c} \bar{\mathbf{s}}_i \log(\hat{\mathbf{s}}_i), \tag{5}$$

where $\mathcal{U}$ denotes the training batch of pixels, which are sampled from the projections of point clouds. Additionally, considering that the semantic categories in the point cloud are heavily unbalanced with vegetation constituting more than 30% while pedestrians constitute less than 1% we employ class-balanced sampling. Here, the sampling probability of each pixel is inversely proportional to the number of point clouds of the category to which that pixel belongs (see Appendix for more details).

**Geometry supervision.** We introduce depth rendering to bolster the model's understanding of geometric information, which can enhance and stabilize pre-training performance. For depth rendering, we adhere to the same method as outlined in Equation 4:

$$\hat{d} = \sum_{i=1}^{N_s} \prod_{j=1}^{i-1} (1 - \alpha_j) \, \alpha_i \delta(\mathbf{r}_i), \tag{6}$$

where $\delta(\mathbf{r}_i)$ denotes the depth of point $\mathbf{r}_i$, and $\alpha_i$ is as defined in Equation 4. During the training phase, we sample batches of pixels projected from the point cloud. The ground-truth depth is determined by the distance between the point and the projection plane. We opt for the view centered on the LiDAR position as the rendering view (referred to as the lidar view henceforth) given that the depth of image views may not be as accurate due to occlusion, as exemplified in Figure 2(b). Finally, we supervise the rendered depth using the L2 loss function:

$$\mathcal{L}_{depth} = \frac{1}{|\mathcal{V}|} \sum_{\mathbf{u} \in \mathcal{V}} \|d_{gt} - \hat{d}\|_2^2, \tag{7}$$

where $d_{gt}$ represents the ground-truth depth, and $\mathcal{V}$ denotes the batch of pixels sampled from the projection plane of the lidar views.

**Point-wise Masking.** Contrary to the patch-wise masking approach utilized in masked signal modeling, we found that point-wise masking with a high masking ratio (95%) performs more effectively within our pre-training framework. Before inputting the point cloud into the encoder, we uniformly sample a subset of the point cloud without replacement and mask the remaining points. Point-wise masking is more proficient at maintaining the semantics of the scene. For smaller objects such as pedestrians, patch-wise masking could potentially obliterate them entirely, leading to a significant loss of semantics. Conversely, point-wise masking tends to make the point cloud of the object more sparse, thus preserving its semantics. Our experimental results further validate the effectiveness of point-wise masking (refer to Section 4.4 for details).

### 3.3 Learning Objective

Following ([50]), we adopt a coarse-to-fine sampling strategy during rendering, utilizing 96 sample points at the coarse stage and 128 sample points at the fine stage. To optimize the model, we minimize the sum of the semantic loss $\mathcal{L}_{semantic}$ and the depth loss $\mathcal{L}_{depth}$, together with the Eikonal term $\mathcal{L}_{reg}$, resulting in a total loss $\mathcal{L}$ defined as follows:

$$\mathcal{L} = \mathcal{L}_{semantic} + \mathcal{L}_{depth} + \lambda \mathcal{L}_{reg}, \tag{8}$$

where $\lambda$ is a hyper-parameter that defaults to 0.1, and $\mathcal{L}_{reg}$ represents the Eikonal term defined by:

$$\mathcal{L}_{reg} = \frac{1}{N_s |\mathcal{V}|} \sum_{\mathbf{u} \in \mathcal{V}} \sum_{i=1}^{N_s} \left( \| \nabla d\left(\mathbf{r}_i\right) \|_2 - 1 \right)^2. \tag{9}$$

While rendering semantics, we apply the stop-gradient to the signed distance in Equation [4]. This is to circumvent the potential unreliability of semantic labels, which could otherwise adversely affect the learning of the scene's geometric information.

Table 1: **Comparisons of 3D object detection performance on the nuScenes validation (top) and test (bottom) sets.** We present mAP, NDS, and AP for each class. Quantitatively, our method surpasses previous state-of-the-art approaches.

| Method | PreTrain | mAP | NDS | Car | Truck | CV. | Bus | Trailer | Barrier | Motor. | Bike | Ped. | TC. |
|---|---|---|---|---|---|---|---|---|---|---|---|---|---|
| CenterPoint ([60]) | ✗ | 56.2 | 64.5 | 84.8 | 53.9 | 16.8 | 67.0 | 35.9 | 64.8 | 55.8 | 36.4 | 83.1 | 63.4 |
| PointContrast ([52]) | ✓ | $56.3_{+0.1}$ | $64.4_{-0.1}$ | - | - | - | - | - | - | - | - | - | - |
| GCC-3D ([22]) | ✓ | $57.3_{+1.1}$ | $65.0_{+0.5}$ | - | - | - | - | - | - | - | - | - | - |
| ProposalContrast ([59]) | ✓ | $57.4_{+1.2}$ | $65.1_{+0.6}$ | 85.0 | 53.8 | 18.5 | 67.2 | 38.5 | 64.9 | 58.1 | 41.7 | 82.5 | 63.7 |
| **Ours** (CenterPoint†) | ✓‡ | $59.0_{+2.8}$ | $66.3_{+1.8}$ | 84.9 | 56.4 | 20.3 | 68.6 | 37.1 | 64.5 | 63.2 | 46.6 | 82.8 | 65.2 |
| GD-MAE ([55]) | ✗ | 58.1 | 65.6 | 85.4 | 56.5 | 16.1 | 70.3 | 36.9 | 64.1 | 59.0 | 39.7 | 84.5 | 68.4 |
|  | ✓ | $58.9_{+0.8}$ | $66.1_{+0.5}$ | 85.4 | 56.8 | 18.5 | 70.1 | 38.2 | 64.4 | 60.9 | 39.3 | 84.7 | 70.4 |
| **Ours** (Second) | ✗ | 52.2 | 63.5 | 84.7 | 56.0 | 13.8 | 68.4 | 35.3 | 57.7 | 45.8 | 18.9 | 79.1 | 61.9 |
|  | ✓‡ | $55.2_{+3.0}$ | $65.5_{+2.0}$ | 84.8 | 59.2 | 17.1 | 71.1 | 38.4 | 58.1 | 55.0 | 23.2 | 80.1 | 64.8 |
| **Ours** (CenterPoint) | ✗ | 61.5 | 68.0 | 85.6 | 60.5 | 20.2 | 71.6 | 37.0 | 66.4 | 64.3 | 49.0 | 86.1 | 74.5 |
|  | ✓‡ | $64.2_{+2.7}$ | $69.7_{+1.7}$ | 85.6 | 60.8 | 25.0 | 72.8 | 40.2 | 67.3 | 71.8 | 58.5 | 85.8 | 73.9 |
| **Ours** (DSVT) | ✗ | 66.4 | 71.1 | **87.8** | 64.1 | 25.4 | **75.9** | 41.9 | 69.2 | 74.4 | 58.5 | 88.1 | 78.8 |
|  | ✓‡ | $68.0_{+1.6}$ | $72.0_{+0.9}$ | 87.6 | **64.7** | 27.5 | 75.6 | **46.9** | 73.8 | **74.7** | **61.0** | **88.5** | 79.5 |
| PointPillars ([20]) | ✗ | 30.5 | 45.3 | 68.4 | 23.0 | 4.1 | 28.2 | 23.4 | 38.9 | 27.4 | 1.1 | 59.7 | 30.8 |
| CBGS ([65]) | ✗ | 52.8 | 63.3 | 81.1 | 48.5 | 10.5 | 54.9 | 42.9 | 65.7 | 51.5 | 22.3 | 80.1 | 70.9 |
| TransFusion ([1]) | ✗ | 65.5 | 70.2 | 86.2 | 56.7 | 28.2 | 66.3 | 58.8 | 78.2 | 68.3 | 44.2 | 86.1 | 82.0 |
| PillarNet-34 ([35]) | ✗ | 66.0 | 71.4 | 87.6 | 57.5 | 27.9 | 63.6 | 63.1 | 77.2 | 70.1 | 42.3 | 87.3 | 83.3 |
| DSVT ([47]) | ✗ | 68.4 | 72.7 | 87.9 | 57.6 | 34.9 | 67.0 | 63.3 | 78.3 | 73.1 | 49.7 | 87.9 | 84.2 |
| **Ours** (CenterPoint) | ✗ | 63.3 | 69.1 | 85.2 | 54.6 | 26.4 | 67.8 | 57.1 | 73.6 | 64.6 | 36.0 | 85.7 | 81.9 |
|  | ✓‡ | $65.9_{+2.6}$ | $70.8_{+1.7}$ | 85.0 | 54.5 | 30.2 | 65.5 | 60.9 | 73.5 | 73.2 | 47.9 | 86.0 | 82.2 |
| **Ours** (DSVT) | ✗ | 68.4 | 72.7 | **87.9** | 57.6 | 34.9 | 67.0 | 63.3 | 78.3 | 73.1 | 49.7 | **87.9** | 84.2 |
|  | ✓‡ | $70.1_{+1.7}$ | $73.7_{+1.0}$ | 87.7 | **58.7** | **38.8** | **68.1** | **65.5** | **79.5** | **76.3** | **53.4** | **87.9** | **84.6** |

† Use 3D sparse convolution network ([60]) as the point cloud encoder.
‡ Use pixel semantics during pretraining.
Notion of class: Construction vehicle (C.V.), pedestrian (Ped.), traffic cone (T.C.).

## 4 Experiments

### 4.1 Experimental Settings

**Dataset: nuScenes** ([4]) is a challenging outdoor dataset providing diverse annotations for various tasks, such as 3D object detection and BEV map segmentation. It comprises approximately 40k keyframes, each equipped with six cameras and a 32-beam LiDAR scan. For 3D object detection, we report the nuScenes detection score (NDS) and mean average precision (mAP). For map segmentation, we provide the mean Intersection over Union (IoU).
**Dataset: ONCE** ([26]) is a large-scale autonomous driving dataset, featuring 1M LiDAR scenes and 7M corresponding camera images. The dataset is split into training, validation, and testing sets consisting of 5k, 3k, and 8k point clouds, respectively. The remaining unannotated point clouds are

Table 2: **3D object detection performance comparisons on the ONCE val split.** We report mAP and AP per class. Our method achieves the best performance among these state-of-the-art methods.

| Methods | PreTrain | PreTrain Data | mAP | Orientation-aware AP | | |
| --- | --- | --- | --- | --- | --- | --- |
| | | | | Vehicle | Pedestrian | Cyclist |
| PointRCNN (38) | ✗ | - | 28.74 | 52.09 | 4.28 | 29.84 |
| PointPillars (20) | ✗ | - | 44.34 | 68.57 | 17.63 | 46.81 |
| SECOND (54) | ✗ | - | 51.89 | 71.19 | 26.44 | 58.04 |
| PV-RCNN (36) | ✗ | - | 53.55 | 77.77 | 23.50 | 59.37 |
| IA-SSD (63) | ✗ | - | 57.43 | 70.30 | 39.82 | 62.17 |
| CenterPoint (60) | ✗ | - | 60.05 | 66.79 | 49.90 | 63.45 |
| DepthContrast (64) | ✗ | - | 51.89 | 71.19 | 26.44 | 58.04 |
| | ✓ | medium | $52.81_{+0.92}$ | $71.92_{+0.73}$ | $29.01_{+2.57}$ | $57.51_{-0.53}$ |
| PointContrast (52) | ✗ | - | 51.89 | 71.19 | 26.44 | 58.04 |
| | ✓ | large | $53.59_{+1.70}$ | $71.87_{+0.68}$ | $28.03_{+1.59}$ | $60.88_{+2.84}$ |
| SLidR (34) | ✗ | - | 28.80 | 52.10 | 4.17 | 30.13 |
| | ✓‡ | large | $30.72_{+1.92}$ | $53.19_{+1.09}$ | $6.74_{+2.57}$ | $32.22_{+2.09}$ |
| ProposalContrast (59) | ✗ | - | 64.24 | 75.26 | 51.65 | 65.79 |
| | ✓ | large | $66.32_{+2.08}$ | $77.22_{+1.96}$ | $54.01_{+2.36}$ | $67.73_{+1.94}$ |
| GD-MAE (55) | ✗ | - | 62.62 | 75.64 | 45.92 | 66.30 |
| | ✓ | large | $64.92_{+2.30}$ | $76.79_{+1.15}$ | $48.84_{+2.92}$ | $69.14_{+2.84}$ |
| **Ours** (Second) | ✗ | - | 52.95 | 74.93 | 24.33 | 59.58 |
| | ✓† | large | $56.10_{+3.15}$ | $77.11_{+2.18}$ | $27.51_{+3.18}$ | $63.69_{+4.11}$ |
| **Ours** (PV-RCNN) | ✗ | - | 54.35 | 78.65 | 24.71 | 59.70 |
| | ✓† | large | $57.45_{+3.10}$ | $80.99_{+2.34}$ | $27.93_{+3.22}$ | $63.44_{+3.74}$ |
| **Ours** (CenterPoint) | ✗ | - | 64.28 | 76.82 | 49.99 | 66.02 |
| | ✓† | small | $65.87_{+1.59}$ | $78.56_{+1.74}$ | $51.50_{+1.51}$ | $67.55_{+1.53}$ |
| | ✓† | medium | $66.72_{+2.44}$ | $79.27_{+2.45}$ | $52.53_{+2.54}$ | $68.36_{+2.34}$ |
| | ✓† | large | $\mathbf{67.41}_{+3.13}$ | $\mathbf{79.50}_{+2.68}$ | $\mathbf{53.36}_{+3.37}$ | $\mathbf{69.36}_{+3.34}$ |

† Use pixel semantics during pretraining.

‡ Use super-pixel and pixel features during pretraining.

divided into three subsets: *small* (100k scenes), *medium* (500k scenes), and *large* (1M scenes) for pre-training purposes. The official evaluation metric is the mean Average Precision (mAP).

**Implementation Details.** We use DSVT-P (47) as our default encoder, given its flexibility. For the decoder, we use a 4-layer MLP with 256 hidden channels, and Softplus is chosen as the activation function. For image segmentation, we utilize DeepLabV3 (7) with MobileNets (18) as the backbone. During pre-training, we train the model using the AdamW (25) optimizer and the one-cycle policy, with a maximum learning rate of $3e^{-4}$. We pre-train the model for 45 epochs on the nuScenes dataset, 20 epochs on the ONCE *small*, 5 epochs on the ONCE *medium*, and 3 epochs on the ONCE *large*. During fine-tuning, we employ random flipping, scaling, rotation, and copy-n-paste as data augmentations, with a maximum learning rate of $3e^{-3}$. All experiments are conducted on NVIDIA V100 GPUs. For more implementation details, please refer to Appendix.

**Overlap of labels between pixel semantics and downstream tasks.** Our pre-training phase utilizes pixel semantic labels that include 19 classes such as road, sidewalk, building, wall, fence, pole, traffic light, sign, vegetation, terrain, sky, person, rider, car, truck, bus, train, motorcycle, and bicycle. In the nuScenes object detection task, the labels include car, truck, construction vehicle, bus, trailer, barrier, motorcycle, bicycle, pedestrian, and traffic cone. For the ONCE object detection task, the labels are limited to vehicle, pedestrian, and cyclist. The nuScenes BEV map segmentation task uses labels such as drivable, pedestrian crossing, walkway, stop line, car park, and divider. There's an overlap in labels like 'car', 'bicycle', and 'pedestrian', and our approach is more in line with the field of weakly supervised learning.

Table 3: **Performance comparisons on nuScenes val BEV map segmentation.** We report mIoU and IoU per class. Our method quantitatively outperforms prior works.

| Method | PreTrain | mIoU | Drivable | Ped. Cross. | Walkway | Stop Line | Carpark | Divider |
|---|---|---|---|---|---|---|---|---|
| PointPillars (20) | ✗ | 43.8 | 72.0 | 43.1 | 53.1 | 29.7 | 27.7 | 37.5 |
| CenterPoint (60) | ✗ | 48.6 | 75.6 | 48.4 | 57.5 | 36.5 | 31.7 | 41.9 |
| DSVT (47) | ✗ | 51.6 | 79.7 | 51.8 | 61.1 | 38.2 | 33.8 | 45.3 |
| **Ours** | ✓ | $\mathbf{55.0_{+3.4}}$ | **82.5** | **54.4** | **65.2** | **40.5** | **39.4** | **48.1** |

Table 4: **Variations in Encoder Types.** We maintain the same components, merely swapping the 3D backbone.

| Encoder | PreTrain | mAP | NDS |
|---|---|---|---|
| 2D Conv (20) | ✗ | 50.0 | 60.7 |
| | ✓ | $53.0_{+3.0}$ | $62.6_{+1.9}$ |
| VoxelNet (60) | ✗ | 56.2 | 64.5 |
| | ✓ | $59.0_{+2.8}$ | $66.3_{+1.8}$ |
| DSVT-P (47) | ✗ | 61.5 | 68.0 |
| | ✓ | $\mathbf{64.2_{+2.7}}$ | $\mathbf{69.7_{+1.7}}$ |

Table 5: **Neural Rendering.** Our rendering-based method outperforms direct projection, showcasing its effectiveness.

| Case | PreTrain | mAP | NDS |
|---|---|---|---|
| baseline | ✗ | 61.5 | 68.0 |
| projection | ✓ | $62.0_{+0.5}$ | $68.3_{+0.3}$ |
| projection-filt | ✓ | $62.8_{+1.3}$ | $68.8_{+0.8}$ |
| rendering w/o depth | ✓ | $63.5_{+2.0}$ | $69.2_{+1.2}$ |
| rendering w/ depth | ✓ | $\mathbf{64.2_{+2.7}}$ | $\mathbf{69.7_{+1.7}}$ |

## 4.2 3D Object Detection

In this section, we investigate pre-training in an autonomous driving context. We benchmark the performance of our model against previous methods on the nuScenes and ONCE datasets.

**NuScenes.** We employ Second (54), CenterPoint (60), and DSVT (47) as our baseline and present the performance on the validation and test sets of nuScenes in Table 1. Our pre-trained model consistently improves the performance of all baselines compared to models trained from scratch. On the validation set, our method boosts the Second, CenterPoint, and DSVT by +3.0/2.0 mAP/NDS, +2.7/1.7 mAP/NDS, and +1.6/0.9 mAP/NDS, respectively. With CenterPoint-Voxel as the detector (5-th row), our method surpasses previous pre-training methods (59; 52; 22) that also utilize CenterPoint-Voxel detector by a large margin. On the test set, our method boosts the CenterPoint and DSVT by +2.6/1.7 mAP/NDS and +1.7/1.0 mAP/NDS, respectively. Moreover, our method significantly outperforms previous state-of-the-art LiDAR detectors (20; 65; 47; 1; 35).

**ONCE.** As shown in Table 2, our model surpasses all other methods on the ONCE validation split. When using CenterPoint (60) as the detector, our method enhances the baseline by 3.13 mAP, outperforming the improvements made by ProposalContrast (59) and GD-MAE (55) by 1.05 and 0.83, respectively. While SLidR (34) also incorporates image information during pre-training, our method significantly outperforms SLidR, showcasing the effectiveness of our approach in utilizing image information. The distinction between the 'no-pretrain' versions of SLidR and our model originates from their foundational detection frameworks, where SLidR employs PointRCNN as its detector. As the scale of pre-training data continues to increase (from *small* to *large*), our method delivers continuous gains, indicating its scalability with respect to data size.

## 4.3 BEV map segmentation

We also demonstrate the versatility of our approach by evaluating its performance on the BEV Map Segmentation task of the nuScenes dataset (4). We provide the IoU results for six background classes in Table 3. Leveraging our PRED, we achieve a significant improvement of 3.4% over DSVT's performance, demonstrating the effectiveness of our approach in map segmentation tasks.

## 4.4 Ablation Studies

In this section, we conduct a series of ablation studies on the nuScenes *val* to explore the essential components of our methodology. Unless otherwise specified, we use CenterPoint (60) as the detector.

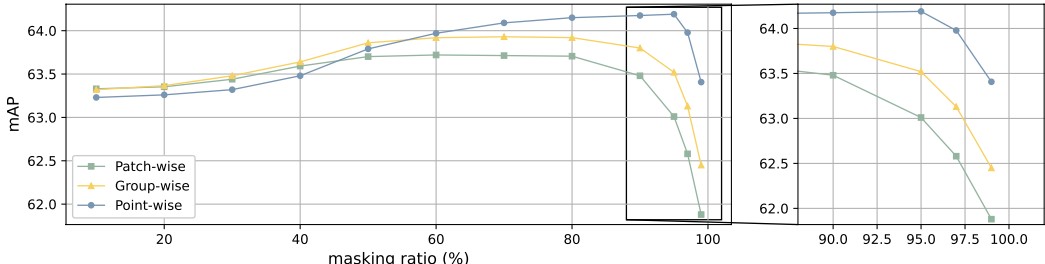

Figure 4: **Masking strategy and masking ratio.** Point-wise masking outperforms patch-wise masking and group-wise masking when a high masking ratio (95%) is applied.

**Encoder type.** To demonstrate the adaptability of our approach across various 3D backbone architectures, we experiment by substituting DSVT-P (47) with 2D Conv from PointPillar (20), and VoxelNet from CenterPoint-Voxel (60). The results, presented in Table 4, show that our approach can boost detector performance by 3.0/1.9 mAP/NDS and 2.8/1.8 mAP/NDS when employing 2D Conv and VoxelNet as backbones, respectively. These improvements are comparable to those achieved using DSVT-P. Due to DSVT-P's superior performance and flexibility, it was chosen as our default encoder. However, our method proves to be effective with a variety of encoder types.

**Semantic rendering design.** Here, we verify the efficacy of our semantic rendering pipeline as shown in Table 5. A basic approach to leveraging images is to directly project the point cloud onto the images. However, as indicated in the 2nd row, this method does not yield significant improvements due to the overlooking of occlusion effects during the projection process. A potential solution is to filter the point cloud based on the image depth, thereby excluding points that do not align with the image depth from the training batch. In the absence of ground-truth image depth, we employ BEVDepth (21) for depth estimation. The enhanced performance in the 3rd row underlines the importance of handling occlusion. However,

Table 6: **Semantic loss.** Both class balance sampling and confidence weight play an important role in calculation of semantic loss.

| Weight | Cls-balance | mAP | NDS |
|--------|-------------|------|------|
|        |             | 62.5 | 68.5 |
| ✓      |             | 63.2 | 69.0 |
|        | ✓           | 63.7 | 69.3 |
| ✓      | ✓           | **64.2** | **69.7** |

the inaccuracy of depth estimation hinders further performance improvement. Our neural rendering approach, shown in the 4th row, surpasses the prior two methods by adaptively managing occlusion, where attributing a reduced weight to occluded points when accumulating semantics as in Equation 4. Ultimately, incorporating depth supervision yields the best results, as depicted in the final row.

Table 6 presents ablation experiments on class balance sampling and confidence weight in semantic supervision. Both components are crucial for semantic supervision; the confidence weight mitigates the effects of inaccurate semantic labels, and class balance sampling addresses the issue of category imbalance in point clouds. For depth supervision, We compared two different rendering views in Table 7. The lidar view outperforms the image view, as the depth of points might be inaccurate in the image view, but not in the lidar view.

Table 7: **LiDAR view.** For depth supervision, the lidar view proves superior to the image view.

| Case | mAP | NDS |
|------|------|------|
| image view | 63.6 | 69.3 |
| lidar view | **64.2** | **69.7** |

**Masking strategy.** In Figure 4, we explore various masking strategies to investigate their impact on our pre-training framework: 1) Patch-wise masking masks a certain fraction of non-empty tokens, with each token representing a non-overlapping patch of the BEV feature map. 2) Group-wise masking masks a certain proportion of point cloud groups, with each group selected using the Furthest Point Sampling (FPS) and $k$-Nearest Neighbor (kNN) algorithm, following (61). 3) Point-wise masking, as elaborated in Section 3.2. As illustrated in Figure 4, group-wise and patch-wise masking strategies perform better when the masking ratio is low. When the ratio reaches 70%, the impact of group-wise and patch-wise masking hits its peak. Intriguingly, the effectiveness of point-wise masking continues to ascend until the masking ratio reaches 95%. The results from point-wise masking surpass those of the other two methods, achieving a score of 64.2 mAP. Consequently, we opt for

the point-wise masking strategy with a 95% masking ratio, as it offers the advantage of accelerated processing while preserving good performance.

**Data-efficient.** A significant advantage of pre-training lies in its ability to enhance data efficiency for downstream tasks that have limited annotated data. In this study, we examine data-efficient 3D object detection by first conducting pre-training on the nuScenes (4) training data, then fine-tuning the pre-trained model with varying fractions of training data: 10%, 20%, 50%, and 100%. The results of these experiments are illustrated in Table 8. Overall, our pre-trained model consistently boosts detection performance, especially when the available labeled data is limited, i.e., improving 4.8 mAP and 3.1 NDS with only 10% labeled data. Remarkably, even when utilizing only 50% of the annotated data, the pre-trained model achieves 63.0 mAP and 68.5 NDS, surpassing the performance of a non-pre-trained version using 100% of the annotated data.

Table 8: **Data Efficiency.** Our approach consistently enhances the detection performance, particularly when the labeled data is limited.

| Dataset fraction | PreTrain | mAP | NDS |
|---|---|---|---|
| 10% | ✗ | 49.6 | 58.9 |
|  | ✓ | $54.4_{+4.8}$ | $62.0_{+3.1}$ |
| 20% | ✗ | 55.1 | 62.9 |
|  | ✓ | $59.3_{+4.2}$ | $65.5_{+2.6}$ |
| 50% | ✗ | 59.7 | 66.2 |
|  | ✓ | $63.0_{+3.3}$ | $68.5_{+2.3}$ |
| 100% | ✗ | 61.5 | 68.0 |
|  | ✓ | $64.2_{+2.7}$ | $69.7_{+1.7}$ |

More experimental results are provided in Appendix.

## 5 Conclusion

In this paper, we introduce PRED, an effective pre-training framework for outdoor point cloud data. Our framework mitigates the inherent incompleteness of point clouds by integrating image information through semantic rendering. It also allows for effective occlusion management by assigning a reduced weight to occluded points during volume rendering. A point-wise masking strategy, with a mask ratio of 95%, is adopted to optimize performance. Extensive experiments validate the effectiveness of our framework as it significantly improves various baselines on the large-scale nuScenes and ONCE datasets across various 3D perception tasks. We hope our PRED can serve as a powerful baseline to inspire future research on point cloud pre-training.

**Limitations.** PRED mainly focuses on point cloud pre-training with images as supervision. The integration of an image-point multi-modality pre-training could potentially improve performance. However, it's beyond this paper's scope and we intend to explore this possibility in our future research.

## Acknowledgements

This work is supported by National Key R&D Program of China (2022ZD0114900) and National Science Foundation of China (NSFC62276005).

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
