# Appendix for "PRED: Pre-training via Semantic Rendering on LiDAR Point Clouds"

**Hao Yang**[1,3]  **Haiyang Wang**[1]  **Di Dai**[2]  **Liwei Wang**[1,2]

[1]Center for Data Science, Peking University

[2]National Key Laboratory of General Artificial Intelligence, School of Intelligence Science and Technology, Peking University    [3]Pazhou Lab

{haoy@stu, wanghaiyang@stu, didai@stu, wanglw@cis}.pku.edu.cn

In the supplementary material, we delve into further details regarding the network architecture and training schemes, which can be found in Section §A. Subsequently, we provide additional experimental results and analysis, illustrated in Sections §B and §C, respectively. For a more visual understanding, please refer to the visualized results in Section §D.

## A  Implementation Details

### A.1  Network Architecture

**Encoder.** Our encoder settings predominantly follow those of DSVT-P (12), a model designed with a structure of four blocks, with each block composed of two DSVT attention layers. For the nuScenes dataset (1), we implement a grid size of (0.3m, 0.3m, 8m). The window size is set at (30, 30, 1), and the maximum number of voxels allocated to each set is 90. Each attention module is equipped with 8 heads, 128 input channels, and 256 hidden channels. Conversely, for the ONCE dataset (10), we employ a grid size of (0.32m, 0.32m, 8m). Here, hybrid window sizes are adjusted to (12, 12, 1) and (24, 24, 1), with the maximum number of voxels per set limited to 36. The attention modules in this case are equipped with 8 heads, 192 input channels, and 384 hidden channels.

**Decoder.** Figure 1 presents a detailed view of our simple fully-connected architecture, which incorporates a four-layer Multilayer Perceptron (MLP) utilizing the Softplus activation function. Positional encoding is applied to spatial location $\mathbf{r}$ with 16 frequencies. In our rendering process, we employ a coarse-to-fine sampling strategy. We initially use 96 sample points during the coarse stage, followed by a finer approach with 128 sample points in the subsequent stage. In accordance with the approach outlined in (13), we maintain a single network. In this configuration, the probability during the coarse sampling stage is calculated based on the S-density function $\Phi_s(d(\mathbf{r}))$, employing fixed standard deviations. On the other hand, the probability for the fine sampling stage is determined based on the same S-density function $\Phi_s(d(\mathbf{r}))$, but this time utilizing the standard deviation $s$ derived from the learning process.

**Class balance sampling.** For a given pixel $\mathbf{u}$ projected from the point clouds, we first determine its semantic category, denoted as $cls$. Subsequently, for each distinct category, we tally the number of pixels that belong to that category, indicated as $N_{cls}$. Ultimately, the sampling probability of pixel $\mathbf{u}$ is formulated to be inversely proportional to $N_{cls}$, which can be mathematically represented as:

$$P(\mathbf{u}) = \frac{1}{N_{cls} \times N_c},  \tag{1}$$

where $N_c$ stands for the total number of categories. With this sampling strategy in place, we aim to achieve a uniform sample distribution across each category during the training process.

37th Conference on Neural Information Processing Systems (NeurIPS 2023).

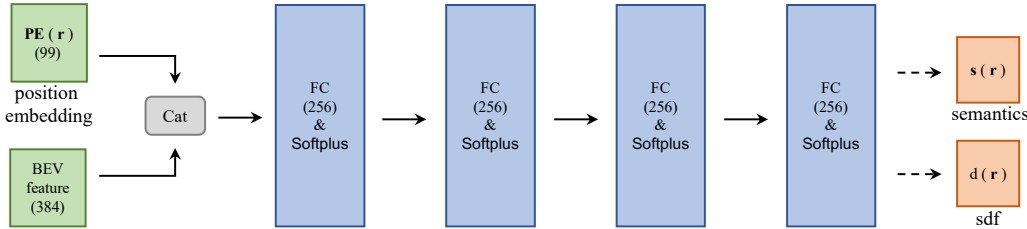

Figure 1: **A visualization of our decoder architecture.** The input vectors are represented in green, while the intermediate hidden layers are displayed in blue, and the output vectors are shown in red. The number embedded in each block indicates the dimension of the respective vector. All layers are designed as fully connected layers, adopting Softplus as the activation function. Initially, we concatenate the positional embedding of the point location, represented by $\mathbf{PE}(\mathbf{r})$, and the Bird's Eye View (BEV) feature, depicted as $\mathbf{F}(\mathbf{r})$. This combined entity is subsequently passed through four fully-connected Softplus layers, each encompassing 256 channels. Ultimately, we derive the predicted semantics $\mathbf{s}(\mathbf{r})$ and the signed distance to the object, denoted by $d(\mathbf{r})$, at position $\mathbf{r}$.

## A.2 Training Schemes

**nuScenes.** For the nuScenes dataset (1), we have designated the detection ranges as (-54.0, 54.0), (-54.0, 54.0), and (-5.0, 3.0). During the pre-training phase, we employ the AdamW optimizer (9), integrating a weight decay of 0.05 and the one-cycle policy, with a maximum learning rate set at 3e-4. The training process is conducted over 45 epochs, utilizing a batch size of 32, and is performed on 8 NVIDIA V100 GPUs. In the fine-tuning phase, we employ data augmentations including random flipping, scaling, rotation, translation, and the copy-n-paste technique. In this stage, the maximum learning rate is set to 3e-3.

**ONCE.** For the ONCE dataset (10), we establish the detection ranges as (-74.88, 74.88), (-74.88, 74.88), and (-5.0, 3.0). During the pre-training phase, we utilize the AdamW optimizer (9), incorporating a weight decay of 0.05 and the one-cycle policy, with a maximum learning rate set at 3e-4. The models are trained using a batch size of 24. Specifically, we conduct 20 epochs on the ONCE *small*, 5 epochs on the ONCE *medium*, and 3 epochs on the ONCE *large*, utilizing 8 NVIDIA V100 GPUs. In the fine-tuning phase, we employ a range of data augmentations, including random flipping, scaling, rotation, and the copy-n-paste technique. For this stage, the maximum learning rate is set to 5e-3.

## B  More Experimental Results

### B.1  More Ablation Studies

**Encoder design.** We performed ablation studies focused on encoder design on the ONCE dataset in order to further substantiate the adaptability of our method to various 3D backbone architectures. In our experiments, we've replaced DSVT-P (12) with 2D Conv from PointPillar (7) and with VoxelNet from CenterPoint-Voxel (18). We use Second (15) as the base detector. The results of these studies can be found in Table 1. These findings underscore the efficacy of our method with diverse encoder designs.

Table 1: **Variations in Encoder Types.** We pre-train encoders on ONCE *large*.

| Encoder | PreTrain | mAP |
|---|---|---|
| 2D Conv (7) | ✗ | 44.34 |
|  | ✓ | $47.41_{+\mathbf{3.07}}$ |
| VoxelNet (18) | ✗ | 51.89 |
|  | ✓ | $55.07_{+\mathbf{3.18}}$ |
| DSVT-P (12) | ✗ | 52.95 |
|  | ✓ | $\mathbf{56.10}_{+\mathbf{3.15}}$ |

**Decoder design.** We conducted an analysis of our decoder design, as illustrated in Tables 2 and 3. Table 2 presents the results of varying the decoder depth, represented by the number of Multilayer Perceptron (MLP) layers. Our results show that a 4-layer MLP rendered optimal performance. During the bird's eye view (BEV) feature extraction process, the positional information of the point cloud undergoes compression, necessitating position embedding to enhance it dur-

Table 2: **Decoder depth.** A reasonably deep decoder works better.

| Layers | mAP | NDS |
|--------|------|------|
| 1 | 63.3 | 69.1 |
| 2 | 63.8 | 69.5 |
| 4 | **64.2** | **69.7** |
| 6 | **64.2** | 69.5 |
| 8 | 63.5 | 69.2 |

Table 3: **Decoder width.** The result is not sensitive to the width.

| Dims | mAP | NDS |
|------|------|------|
| 64 | 63.7 | 69.4 |
| 128 | 64.0 | 69.5 |
| 256 | **64.2** | **69.7** |
| 512 | **64.2** | **69.7** |
| 1024 | 63.9 | 69.4 |

Table 5: **Image Segmentation Models.**

| Image Segmenter | PreTrain | mAP | NDS |
|-----------------|----------|------|------|
| Baseline | ✗ | 61.5 | 68.0 |
| PSPNet (19) | ✓ | $63.9_{+2.4}$ | $69.5_{+1.5}$ |
| DeepLabv3 (MobileNet) (3) | ✓ | $64.2_{+2.7}$ | $69.7_{+1.7}$ |
| DeepLabv3 (ResNet101) (3) | ✓ | $64.4_{+2.9}$ | $69.7_{+1.7}$ |
| SegFormer (14) | ✓ | $64.3_{+2.8}$ | $\mathbf{69.9}_{+1.9}$ |
| Semantic-Segment-Anything (2) | ✓ | $\mathbf{64.5}_{+3.0}$ | $\mathbf{69.9}_{+1.9}$ |

ing decoding. If the depth is insufficient, the decoder may not adequately integrate the BEV feature and position embedding information, potentially undermining the pre-training effect. Consequently, a decoder with reasonable depth proves more effective.

Table 3 displays the results derived from altering the decoder width, which corresponds to the number of channels. We observed that the performance remains relatively insensitive to the width. Thus, we opted for a default width of 256-d, which offers strong performance and reduced computational demand. In sum, our default decoder is both lightweight and efficient, boasting 4 MLP layers and a width of 256-d.

**Coarse-to-fine sampling strategy.** We present the impact of the coarse-to-fine sampling strategy on our method in Table 4. The table illustrates that the coarse-to-fine sampling strategy outperforms the coarse sampling strategy. This improvement is attributed to the fact that the coarse-to-fine approach facilitates more precise sampling in larger scenes, thereby enhancing the accuracy of the rendering process.

Table 4: **Coarse-to-fine sampling strategy.**

| Case | mAP | NDS |
|------|------|------|
| coarse only | 63.4 | 69.1 |
| coarse-fine | **64.2** | **69.7** |

**Image Segmentation Model Choices.** Here we conducted ablation studies involving a range of renowned segmentation models, namely PSPNet (19), DeepLabv3 (3) (MobileNet (5)), DeepLabv3 (ResNet101 (4)), and SegFormer (14). Keeping other settings consistent with Section 4.4, our results (as presented in Table 5) demonstrated robustness in pre-training performance across these models. We have also tried to substitute the segmenter with a SAM (6)-based approach, named Semantic-Segment-Anything (SSA) (2). The results are outlined in Table 5. Thanks to SAM's strong generalization capabilities and segmentation performance, our method demonstrated further enhancements when paired with SSA as the segmenter.

## B.2 Waymo Open Dataset

The Waymo Open Dataset (WOD) (11) is a widely utilized benchmark for outdoor 3D perception, encompassing 1150 point cloud sequences in total, amounting to over 200K frames. Each frame offers a vast perception range of 150m Œ 150m. All outcomes are assessed by the standard protocol using 3D mean Average Precision (mAP) and its weighted variant by heading accuracy (mAPH).

**Experimental Settings.** Our methodology primarily aligns with the settings of DSVT-P (12), constructed with four blocks, each containing two DSVT attention layers. All attention modules are

Table 6: **3D object detection performance comparisons on Waymo Open validation set with 20% labeled data (top) and 100% labeled data (bottom).** We report AP and APH per class under the LEVEL 2 metric.

| Methods | PreTrain | Overall | | Vehicle | | Pedestrian | | Cyclist | |
|---|---|---|---|---|---|---|---|---|---|
| | | AP | APH | AP | APH | AP | APH | AP | APH |
| GCC-3D (8) | ✗ | 63.5 | 61.0 | 61.8 | 61.3 | 63.6 | 57.8 | 65.0 | 63.8 |
| | ✓ | $65.3_{+1.8}$ | $62.8_{+1.8}$ | 64.0 | 63.5 | 64.2 | 58.5 | 67.7 | 66.4 |
| ProposalContrast (17) | ✗ | 63.5 | 61.0 | 61.8 | 61.3 | 63.6 | 57.8 | 65.0 | 63.8 |
| | ✓ | $66.4_{+2.9}$ | $63.9_{+2.9}$ | 64.9 | 64.4 | 66.1 | 60.1 | 68.2 | 67.0 |
| GD-MAE (16) | ✗ | - | 65.4 | - | - | - | - | - | - |
| | ✓ | 70.2 | $67.1_{+1.7}$ | 67.7 | 67.2 | 73.2 | 65.5 | 69.9 | 68.7 |
| **Ours** (DSVT) | ✗ | 70.7 | 68.4 | 68.7 | 68.5 | 74.1 | 68.2 | 69.4 | 68.6 |
| | ✓ | $73.2_{+2.5}$ | $70.8_{+2.4}$ | **70.7** | **70.1** | **75.1** | **69.6** | **73.7** | **72.5** |
| GD-MAE (16) | ✗ | - | 67.3 | - | - | - | - | - | - |
| | ✓ | 70.6 | $67.6_{+0.3}$ | 68.7 | 68.3 | 72.8 | 65.5 | 70.3 | 69.2 |
| **Ours** (DSVT) | ✗ | 73.2 | 71.0 | 70.9 | 70.5 | 75.2 | 69.8 | 73.6 | 72.7 |
| | ✓ | $74.0_{+0.8}$ | $71.8_{+0.8}$ | **71.5** | **71.2** | **75.9** | **70.5** | **74.5** | **73.6** |

equipped with 8 heads, 192 input channels, and 384 hidden channels. The encoder's hybrid window sizes are set to (12, 12, 1) and (24, 24, 1), with a maximum of 36 voxels allocated to each set. In the context of the Waymo (11) dataset, the detection ranges are determined as (-74.88, 74.88), (-74.88, 74.88), and (-2.0, 4.0). During the pre-training phase, we train the model with 100% of the data, using the AdamW (9) optimizer with a weight decay of 0.05 and the one-cycle policy, with a maximum learning rate of 3e-4. The models are trained using a batch size of 24 across 30 epochs on 8 NVIDIA V100 GPUs. During the fine-tuning phase, we incorporate data augmentations including random flipping, scaling, rotation, translation, and the copy-n-paste technique, with a maximum learning rate set at 3e-3.

**Results.** We leverage DSVT (12) as our baseline and illustrate its performance on the validation sets of the Waymo dataset using single-frame LiDAR input in Table 6. When fine-tuning with 20% labeled data, our method enhances the performance by +2.5/2.4 AP/APH, thereby demonstrating our method's efficacy on the Waymo dataset. When fine-tuning with 100% labeled data, our method continues to boost performance, achieving final scores of 74.0/71.8 AP/APH. Compared to the results on nuScenes (1) and ONCE (10), the improvement on the Waymo dataset is somewhat smaller. This can be attributed to two main factors: firstly, the images in Waymo only cover front and side perspectives, hence only a portion of the scene can be trained during the pre-training phase. Secondly, Waymo's point cloud data is more comprehensive, meaning the ambiguity of the target has relatively less influence on the training process. In general, our method still performs well on the Waymo dataset.

## C   Ambiguity in Contrastive Learning

In Figure 2, we underscore how the incompleteness of point clouds can introduce ambiguity into the contrastive learning process. This ambiguity, in turn, may have a detrimental effect on the training process.

## D   Qualitative Results

### D.1   Predicted Semantic Maps

In the pre-training phase, we only select a minor fraction of the image pixels as training samples, which are projected from point clouds, representing roughly 5% of the total pixel count. Remarkably, we've observed that our model is capable of extrapolating a full semantic map from the point cloud,

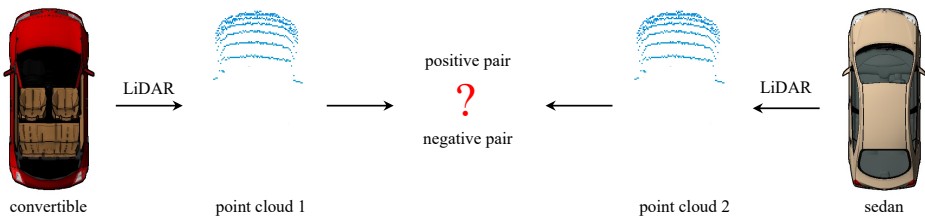

Figure 2: **Ambiguity in contrastive learning.** The image features a convertible car and a sedan displayed on the left and right sides respectively, constituting a negative pair in the contrastive learning scenario. However, due to the incomplete data obtained after LiDAR scanning, their point cloud representations become strikingly similar, injecting ambiguity into contrastive learning process.

as demonstrated in Figure 3. The predicted semantic map underscores an intriguing revelation. Despite the point cloud information being incomplete, our pre-training methodology enables the model to derive a more comprehensive representation of the point cloud.

## D.2    Predicted Point Clouds Semantics

We demonstrate the semantics predicted for each point during the pre-training phase in Figure 4. To predict the semantics for point $\mathbf{r}$, we input its position embedding along with the corresponding BEV feature into the decoder, as illustrated in Figure 1. Interestingly, our pre-training approach of semantic rendering enables robust prediction of the 3D point cloud semantics even in the absence of 3D scene annotations.

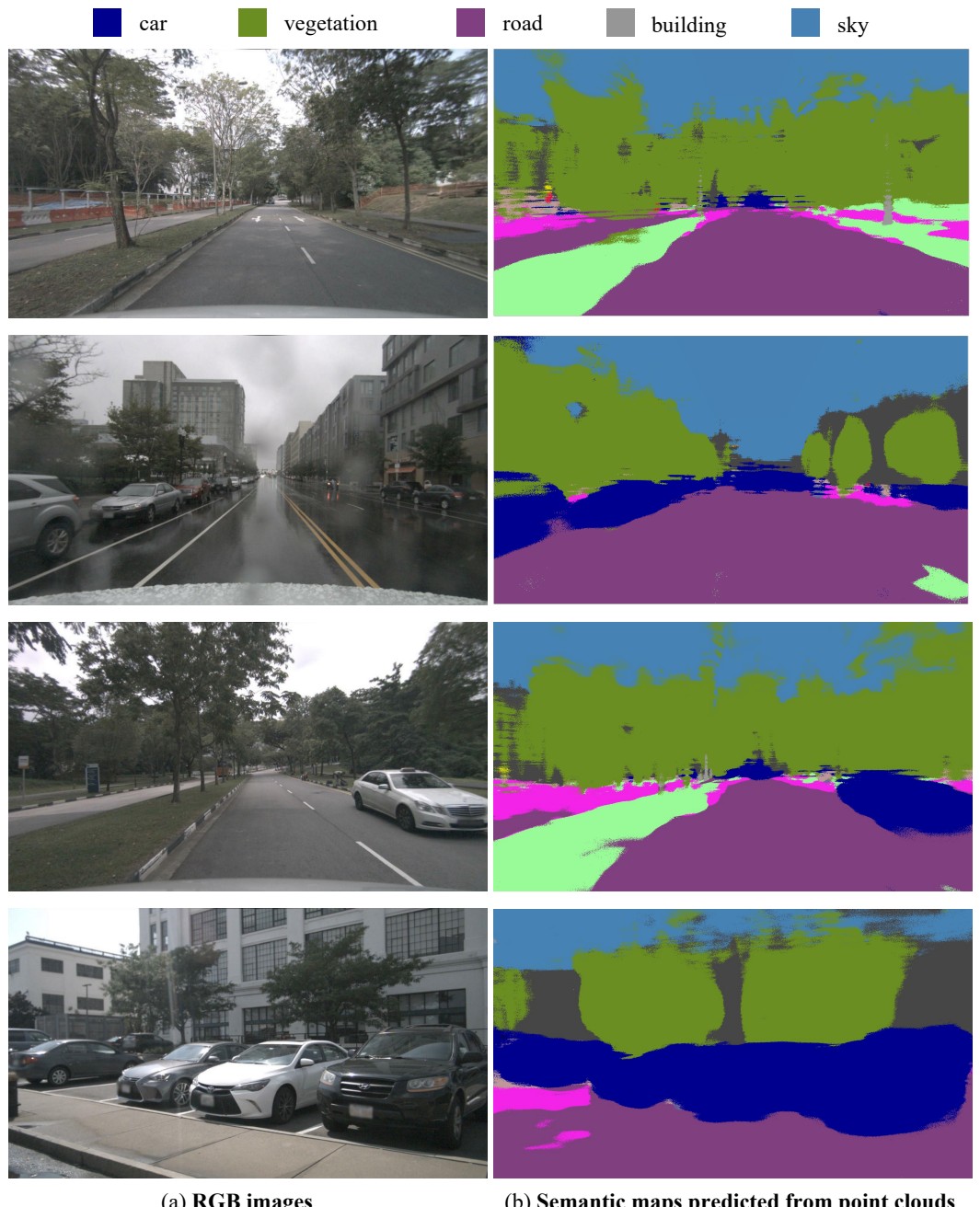

| car | vegetation | road | building | sky |

(a) **RGB images**  (b) **Semantic maps predicted from point clouds**

Figure 3: **An illustrative depiction of the semantic maps predicted from LiDAR point clouds during pre-training.** On the left, we exhibit RGB images captured by the camera in varied scenes, while the corresponding semantic maps, as predicted by the model from the point clouds during pre-training, are displayed on the right. Notably, despite the incomplete nature of the point cloud data, our model successfully generates comprehensive semantic maps.

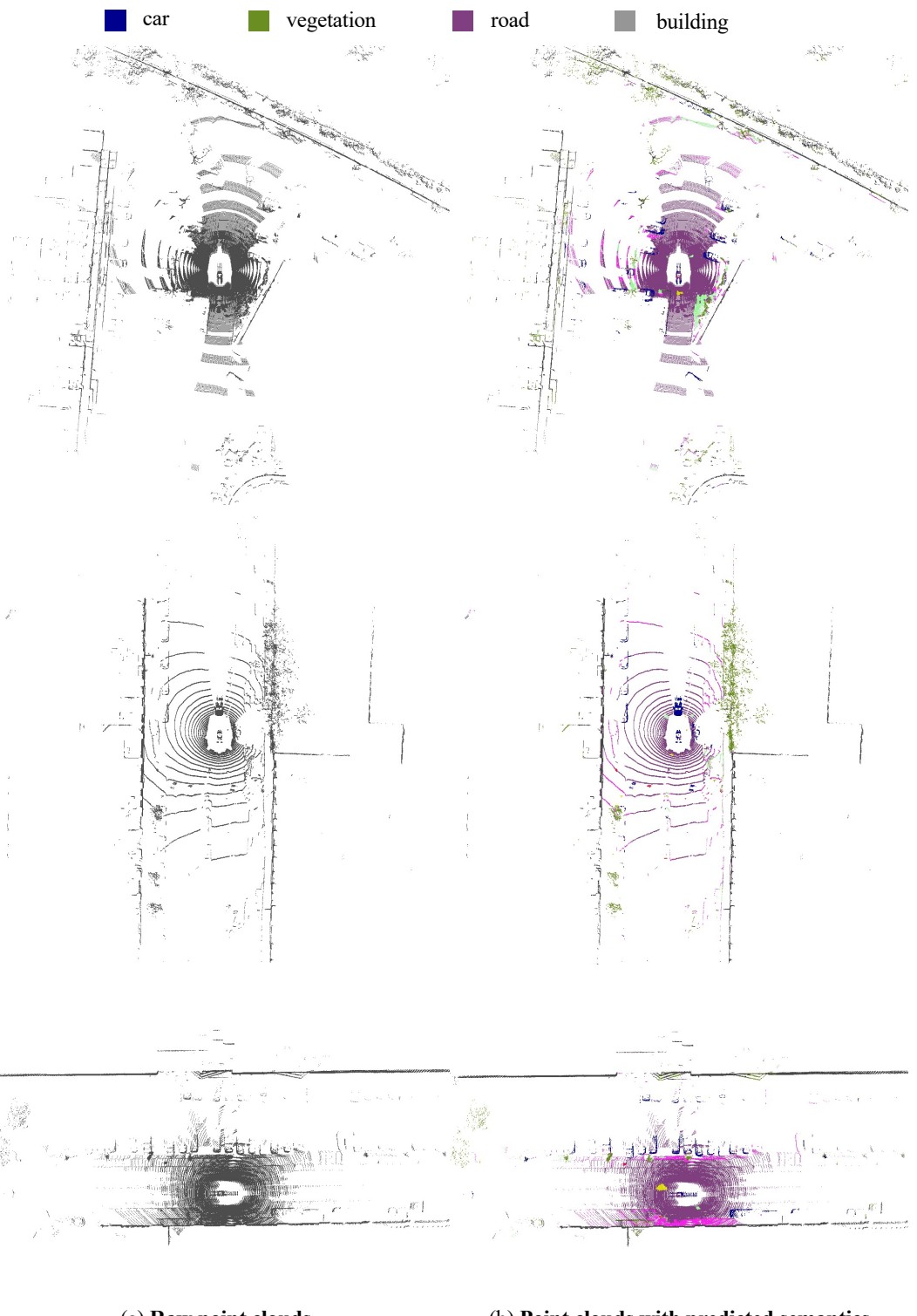

(a) **Raw point clouds.**  (b) **Point clouds with predicted semantics.**

Figure 4: **An illustration of the predicted semantics of point clouds.** The raw point cloud is depicted on the left, while the corresponding point cloud augmented with predicted semantics is showcased on the right. Notably, during the pre-training stage, our model exhibits a proficient capability in predicting the semantics of the 3D point cloud, even in the absence of 3D scene annotations.