# OpenReview forum: "PRED: Pre-training via Semantic Rendering on LiDAR Point Clouds"
_NeurIPS.cc/2023/Conference — NeurIPS 2023 poster_

### Official Review · Reviewer_sbFA · 2023-07-04

**Soundness:** 3 good
**Presentation:** 4 excellent
**Contribution:** 2 fair
**Rating:** 5
**Confidence:** 5

**Summary:**

This work incorporates images into point cloud pretraining since images contain richer semantic information. Instead of adopting the back propagation strategy which can not handle the misalignment between camera and LiDAR, this paper leverages the neural rendering technique to injecting the semantics into the representation learning process.

**Strengths:**

(1) This paper is well-written and presents a clear and concise idea.
(2) The motivation of leveraging the neural rendering to overcome the mismatch of pixel and point is compelling.
(3) The experimental results demonstrate consistent improvement on different datasets with varying baselines. And the ablation studies validate the contribution of each component in the pipeline.

**Weaknesses:**

(1) The technical contribution is limited. For example, the BEV conditioned semantic neural rendering strategy, and the masking strategy have been studied in previous works.
(2) Although the occlusion problem can be alleviated by assigning a reduced weight to occluded points, the rendering process will introduce additional noise since it assigns multiple semantic labels to each point, leading to the semantic ambiguity.
(3) The rendering process allow every points in the ray receive the gradients which leads to many irrelevant points being optimized.
(4) The authors claims that the neural rendering is superior to the point-to-pixel projection, but there is no performance comparison between these two methods in the experiments. I think the latter may also bring an obvious improvement since the inambiguous projections.
(5) This work takes the semantic labels as supervision for neural rendering. I think it would be better to use the 2D feature vectors as the supervision.

**Questions:**

My major concern lies in the superiority of this rendering process compared to the point painting process. While the author address the occlusion issue, I firmly believe that it can be effectively resolved by accurately determining the object boundaries. By precisely locating the boundaries of objects, the occlusion challenge can be easily mitigated, potentially diminishing the need for a complex rendering approach.

**Limitations:**

The author address the limitations.

---

> ### Author Rebuttal · Authors · 2023-08-09
>
> We appreciate your positive feedback regarding the paper's clarity, motivation, and experimental results. We understand your concerns and would like to address your points one by one.
>
> **Q1: Concerns regarding technical contribution.**
>
> **A1:** Our main contribution lies in a novel pre-training framework for outdoor point clouds. By combining neural rendering with point-wise masking and using 2D semantic labels as supervision, we provide a comprehensive solution that effectively addresses the challenges of reconstruction ambiguity and occlusion in point cloud pre-training. The effectiveness of our approach has been demonstrated through improved performance on several downstream tasks. Moreover, to the best of our knowledge, we are the first to introduce a BEV-conditioned semantic neural rendering strategy for point cloud pre-training.
>
> ---
>
> **Q2: Semantic ambiguity arising from the rendering process.**
>
> **A2:** Assigning multiple labels to a single point can be a source of noise. However, our dataset's emphasis on autonomous driving scenarios mitigates this concern. Given the car-centric viewpoint of our images, overlaps between different perspectives remain minimal. For instance, in the nuScenes dataset, the overlap between two views is less than 10%. Consequently, multi-label rendering affects a limited set of points. Future research can further mitigate this by selectively sampling non-overlapping regions, a refinement we'll underscore in our revision.
>
> ---
>
> **Q3: The rendering process allows every point in the ray to receive the gradients which leads to many irrelevant points being optimized.**
>
> **A3:** The rendering process allows for the geometric understanding of the scene. When rendering semantics, we apply the stop-gradient to the signed distance as explained in Line 162. This means that only a small number of significant points with larger weights are primarily optimized by semantic loss. In our depth rendering, while we don't curtail gradients to seemingly 'irrelevant' points, these contribute substantially to the model's geometric understanding of the scene. This geometry perception proves pivotal for downstream tasks, like object detection.
>
> ---
>
> **Q4: Contrasting neural rendering with point-to-pixel projection.**
>
> **A4:** Sorry for any oversight. Kindly direct your attention to Table 5 for a comprehensive comparison between neural rendering and point-to-pixel projection. The results illustrate the substantial superiority of our neural rendering approach over the point-to-pixel projection method.
>
> ---
>
> **Q5: Embracing 2D feature vectors as supervision.**
>
> **A5:** We have explored the option of supervising the model with feature vectors extracted from the segmenter's backbone. Through comparative results (shown below) under the experimental settings as detailed in Section 4.4, we find that both supervision signals are effective.
>
> | Supervision       | PreTrain | mAP      | NDS      |
> |:-----------------:|:--------:|:--------:|:--------:|
> | baseline          | ❌       | 61.5     | 68.0     |
> | semantic label    | ✔️       | 64.2$_{+2.7}$ | 69.7$_{+1.7}$ |
> | feature vector    | ✔️       | 64.3$_{+2.8}$ | 69.4$_{+1.4}$ |
>
> We ultimately chose to use supervision from semantic labels as it simplifies the computation of the loss function. Specifically, semantic labels are utilized for class-balanced sampling and loss weights, as outlined in Lines 128-135.
>
> Nonetheless, the potential of 2D feature vectors remains promise, particularly when sourced from advanced Vision Foundation Models such as SAM [60]. Even though SAM doesn't predict semantic labels, its impressive generalization capabilities could enhance the effectiveness of our method—a prospect worth deeper exploration.
>
> [60] SAM: Segment Anything.
>
> ---
>
> **Q6: Comparing neural rendering to the point painting process.**
>
> **A6:** The idea of utilizing accurate object boundaries to address occlusion challenges is indeed intriguing. However, in the context of autonomous driving, which involves sparse and incomplete point clouds, accurately determining boundaries becomes challenging due to these constraints. Furthermore, this methodology would require pre-processing object segmentation and 3D boundary computations, potentially increasing system complexity. While we value your perspective, our research leverages neural rendering, effectively managing occlusion without resorting to explicit boundary detection. Nevertheless, exploiting the inductive bias of point clouds to handle occlusion presents a promising direction that we are eager to explore in future efforts.

---

> > ### Comment · Reviewer_sbFA · 2023-08-20
> >
> > After carefully reading other comments, I believe the solution proposed by the author is fancy but lack of substance. Although the experiments validate the rendering-based approach over point-to-pixel approach, the experimental justification only relying on a few metrics of downstream tasks appears to be rather weak. I am now inclined to give 5 (broadline accept).

---

> > > ### Author Response · Authors · 2023-08-20
> > > **Thanks for your feedback**
> > >
> > > Thanks for your comments. In this work, we present a semantic rendering approach for point cloud pre-training. This approach effectively addresses challenges related to reconstruction ambiguities and occlusions. Our method consistently demonstrates enhancements compared to various baseline methods across a diverse range of datasets and tasks. We will further revise the paper accordingly.

---

### Official Review · Reviewer_73N2 · 2023-07-06

**Soundness:** 3 good
**Presentation:** 2 fair
**Contribution:** 4 excellent
**Rating:** 8
**Confidence:** 4

**Summary:**

This paper investigates weakly-supervised representation learning for outdoor LiDAR point clouds. To start, the authors point out that the inherent incompleteness of outdoor LiDAR points would reduce the effectiveness of self-supervised representation learning approaches. To mitigate this, the authors propose to use synchronized images as additional signals to supervise the representation learning process. Observing the lack of color information in the point cloud and the mismatch between points and pixels due to occlusion, the authors propose to use neural rendering of pixel semantics as a pre-text task to circumvent the two problems. Specifically, with a slight modification to NEUS’s weighting function, the authors build an implicit neural representation on top of the BEV LiDAR features, positing that good BEV LiDAR features for implicit neural representations could be good representations for any downstream recognition tasks. Extensive experiments have been conducted to show the effectiveness of the approach.

**Strengths:**

- Novelty - Using neural rendering as a pre-text task for point cloud representation learning is a novel idea the reviewer has never seen in the literature.
- Beautiful figures - Figure 1,2,3 are aesthetically pleasing, with figure 2 clearly illustrating the core problems in representation learning for outdoor LiDAR point clouds.
- Strong performance improvement over baselines that do not use images as additional signals.
- Thorough experiments - Experiments thoroughly demonstrate the strength of the approach.

**Weaknesses:**

Overall, this is a strong paper but the reviewer has to point out three weaknesses — two related to presentation and the other related to baselines.

- (Presentation 1) One of the contributions of the paper is to show that images could be valuable signals for pre-training representation. However, it is unclear which baselines in table 1 and 2 actually use additional images as signals. Also, it is unclear what type of signals are being used (i.e. raw pixel values vs pixel semantics).
- (Baseline) Another contribution of the paper is to show that pixel semantics is a useful signal for pre-training representation. Although the proposed approach did not converge when trying to render raw pixel value (line 101-103), SLidR (table 2) is a baseline that actually leverages color information to form superpixel for pre-training. Given the big difference between the no-pretrain variants for SLidR and ours, it is difficult to judge whether pixel-value is a weaker signal or there is some other difference (such as model architecture or optimization recipes) that is causing SLidR to underperform PRED.
- (Presentation 2): Since using pixel semantics is part of the core contributions, it is important for the authors to acknowledge/mention whether there are any overlaps between the pixel semantic label space and the downstream task label space. Without this, it is difficult to tell whether the proposed approach is a self-supervised approach or a weakly-supervised approach.

**Questions:**

The reviewer believes this is a good paper. However, the reviewer thinks it is important to properly address the weaknesses (especially on the SLiDR baseline and presentation 2). For pre-rebuttal, the reviewer would give “borderline reject” to implore the authors to properly address the weaknesses. If the weaknesses are sufficiently addressed, the reviewer is more than happy to give this submission a “strong accept”.

Suggestions:
- To fix the first weakness in the presentation, the reviewer recommends adding a column in table 1 and 2 to indicate whether and the type of pixel signal used. Also, it would be great to emphasize this in section 4.2.
- To fix the second weakness in the presentation, the reviewer recommends adding a paragraph indicating label overlap between the pixel semantics and downstream tasks in section 4.1.
- For the baseline, the reviewer would like to know why the “no-pretrain” SLiDR baseline is much worse than the "no-pretrain" OURS baseline is table 2.

**Limitations:**

Yes, the authors have adequately addressed the limitations.

---

> ### Author Rebuttal · Authors · 2023-08-09
>
> We are encouraged by your acknowledgment of our work's novelty and thoroughness in experimentation. We aim to address the concerns raised to offer clearer insights into our methodology.
>
> **Q1: Clarity on image signals in Tables 1 and 2.**
>
> **A1:** To offer greater clarity, we will introduce a dedicated column in Tables 1 and 2 that indicates which baselines utilize image signals and the specific type of those signals - be it raw pixel values or pixel semantics.
>
> Here are rough versions of how the updated tables might look:
>
> Table 1:
>
> | Method           | PreTrain | Pixel Signal Used | mAP      | NDS      |
> |:----------------:|:--------:|:-----------------:|:--------:|:--------:|
> | CenterPoint      | ❌       | -                 | 56.2     | 64.5     |
> | PointContrast    | ✔️       | -                 | 56.3$_{+0.1}$ | 64.4$_{−0.1}$ |
> | GCC-3D           | ✔️       | -                 | 57.3$_{+1.1}$ | 65.0$_{+0.5}$ |
> | ProposalContrast | ✔️       | -                 | 57.4$_{+1.2}$ | 65.1$_{+0.6}$ |
> | GD-MAE           | ❌       | -                 | 58.1     | 65.6     |
> | GD-MAE           | ✔️       | -                 | 58.9$_{+0.8}$ | 66.1$_{+0.5}$ |
> | Ours (CenterPoint) | ❌       | -   | 61.5 | 68.0 |
> | Ours (CenterPoint) | ✔️       | pixel semantics   | 64.2$_{+2.7}$ | 69.7$_{+1.7}$ |
>
>
> Table 2:
>
> | Method              | PreTrain | Pixel Signal Used        | mAP      |
> |:-------------------:|:--------:|:------------------------:|:--------:|
> | PointRCNN           | ❌       | -                        | 28.74    |
> | SECOND              | ❌       | -                        | 51.89    |
> | CenterPoint         | ❌       | -                        | 60.05    |
> | PointContrast       | ❌       | -                        | 51.89    |
> | PointContrast       | ✔️       | -                        | 53.59$_{+1.70}$ |
> | SLidR               | ❌       | - | 28.80    |
> | SLidR               | ✔️       | super-pixel & pixel feature | 30.72$_{+1.92}$ |
> | ProposalContrast    | ❌       | -                        | 64.24    |
> | ProposalContrast    | ✔️       | -                        | 66.32$_{+2.08}$ |
> | GD-MAE              | ❌       | -                        | 62.62    |
> | GD-MAE              | ✔️       | -                        | 64.92$_{+2.30}$ |
> | Ours (CenterPoint)  | ❌       | -           | 64.28    |
> | Ours (CenterPoint)  | ✔️       | pixel semantics           | 67.41$_{+3.13}$ |
>
> Additionally, we'll underscore the significance of harnessing image signals in Section 4.2, elaborating how distinct pixel signals like raw pixel values and pixel semantics can profoundly affect model efficacy.
>
> ---
>
> **Q2: Performance discrepancy between 'no-pretrain' SLiDR and OURS baselines in Table 2.**
>
> **A2:** The distinction between the 'no-pretrain' versions of SLidR and our model originates from their foundational detection frameworks. SLidR employs PointRCNN as its detector, a methodology grounded in point-based detection. In contrast, our approach is designed for a BEV-based detector, which typically yields a superior baseline performance. Even if our baseline has less room for improvement, the boost our method offers over the baseline surpasses that of SLidR. For instance, our method (CenterPoint) achieves a remarkable +3.13mAP improvement compared to SLidR's +1.92mAP gain.
>
> One more clarification, SLidR's methodology extends beyond the utilization of color information; it incorporates image features for point cloud contrastive learning supervision. These image features are extracted through a pre-trained ResNet employing MoCov2. In contrast, our strategy only uses pixel semantics.
>
> We will clarify these points in the revised version of our paper.
>
> ---
>
> **Q3: Overlap of labels between pixel semantics and downstream tasks.**
>
> **A3:** In our revised manuscript's Section 4.1, we will incorporate a detailed overview indicating the label overlaps between pixel semantics used in our pre-training and those of our downstream tasks.
>
> Here is a preliminary look at the label overlap:
>
> 1. Pixel Semantics: Our pre-training phase utilizes pixel semantic labels that include 19 classes such as road, sidewalk, building, wall, fence, pole, traffic light, sign, vegetation, terrain, sky, person, rider, car, truck, bus, train, motorcycle, and bicycle.
> 2. Downstream tasks:
>    - In the nuScenes object detection task, the labels include car, truck, construction vehicle, bus, trailer, barrier, motorcycle, bicycle, pedestrian, and traffic cone.
>    - For the ONCE object detection task, the labels are limited to vehicle, pedestrian, and cyclist.
>    - The nuScenes BEV map segmentation task uses labels such as drivable, pedestrian crossing, walkway, stop line, car park, and divider.
>
> There's an overlap in labels like 'car', 'bicycle', and 'pedestrian', and our approach is more in line with the field of weakly supervised learning.
>
> Furthermore, we have the prospect of integrating Vision Foundation Models (VFMs), like SAM [60], into our framework, capitalizing on their semantic features as the supervision. These models exhibit enhanced generalization capabilities, which could potentially result in further performance improvements for our method. However, given the relatively brief existence of VFMs, there remains some ambiguity regarding whether pre-training with VFMs falls within the realm of self-supervised or weakly-supervised methods. We will discuss these in the revision.
>
> [60] SAM: Segment Anything.

---

> > ### Comment · Reviewer_73N2 · 2023-08-20
> > **Response**
> >
> > The reviewer thanks the authors for the detailed response. The reviewer's concerns are sufficiently addressed and decides to raise the rating to strong accept.

---

> > > ### Author Response · Authors · 2023-08-20
> > > **Thanks for your positive feedback!**
> > >
> > > Thank you very much for the positive feedback!
> > >
> > > Your constructive comments and suggestions are very helpful in improving our paper quality. Thanks!

---

### Official Review · Reviewer_8P8s · 2023-07-06

**Soundness:** 2 fair
**Presentation:** 3 good
**Contribution:** 3 good
**Rating:** 5
**Confidence:** 4

**Summary:**

This paper proposed a novel point cloud pre-training framework, PRED, which leverages the semantic information consistency between the LiDAR point clouds and the camera images to improve the point cloud pre-training performance. The author proposed (1) a novel semantic rendering module for decoding the semantics from the BEV feature maps and (2) a point-wise masking mechanism to alleviate the reconstruction ambiguity.

The proposed pre-training method demonstrates superior performance according to the experiments section.

**Strengths:**

(1), To the best of my knowledge, the proposed point cloud pre-training framework with semantic rendering is novel and reasonable.
(2), point cloud pre-training is an important task for academia and industry.
(3), According to the experiment section, the proposed framework PRED has achieved superior performance on multiple benchmarks.

**Weaknesses:**

(1), Dealing with occlusion is claimed as one of this paper's major contributions and advantages. However, why the occluded points will be allocated a lower weight is not illustrated clearly in the paper. The author could add more presentation, analysis, visualization, and evaluation.
(2), The pre-trained semantic model is required for the proposed approach. This point should be marked and compared with other methods in Tables 1 and 2.

**Questions:**

(1), The proposed method relies on well-trained 2D segmentation models, which limits its generalization ability.

**Limitations:**

(1) Is there anything special or novel for handling occlusion compared to [46]? If not, then this point should not be highlighted.
(2) How would the proposed method perform if the 2D segmenter fails?

---

> ### Author Rebuttal · Authors · 2023-08-09
>
> The recognition of our approach's potential significance in both academic and industrial circles is particularly encouraging. We acknowledge the concerns you've highlighted and would like to offer clarifications:
>
> **Q1: Is there anything special or novel for handling occlusion compared to [46]? Why the occluded points will be allocated a lower weight?**
>
> **A1:** Sorry for any confusion. The key point we aim to emphasize in our paper is the importance of addressing occlusion during point cloud pre-training, a factor that has been overlooked in prior works. In this regard, we apply neural rendering, drawing inspiration from and aligning with [46], to effectively handle occlusion with encouraging results.
>
> The allocation of a lower weight to occluded points stems from our volume rendering computation, as outlined in Equation 2. For two depth values $t_1$ and $t_2$ that satisfy $d(t_1) = d(t_2)$ and $t_1 < t_2$, we have
> $$w(t_1) - w(t_2) = \exp \left(-\int_0^{t_1} \rho(u) \mathrm{d} u\right) \rho(t_1) - \exp \left(-\int_0^{t_2} \rho(u) \mathrm{d} u\right) \rho(t_2) = (\exp \left(-\int_0^{t_1} \rho(u) \mathrm{d} u\right) - \exp \left(-\int_0^{t_2} \rho(u) \mathrm{d} u\right)) \rho(t_1) > 0.$$
> This indicates that, for points with equivalent SDF values, those proximal to the viewpoint are awarded greater weight, resulting in occluded points receiving reduced weight during rendering. We will clarify this in the revised manuscript.
>
> ---
>
> **Q2: The pre-trained semantic model is required for the proposed approach, which should be marked and compared with other methods in Tables 1 and 2.**
>
> **A2:** Your suggestion is well-received. We will address this point in Tables 1 and 2.
>
> Here are rough versions of how the updated tables might look:
>
> Table 1:
>
> | Method           | PreTrain | Pixel Signal Used | mAP      | NDS      |
> |:----------------:|:--------:|:-----------------:|:--------:|:--------:|
> | CenterPoint      | ❌       | -                 | 56.2     | 64.5     |
> | PointContrast    | ✔️       | -                 | 56.3$_{+0.1}$ | 64.4$_{−0.1}$ |
> | GCC-3D           | ✔️       | -                 | 57.3$_{+1.1}$ | 65.0$_{+0.5}$ |
> | ProposalContrast | ✔️       | -                 | 57.4$_{+1.2}$ | 65.1$_{+0.6}$ |
> | GD-MAE           | ❌       | -                 | 58.1     | 65.6     |
> | GD-MAE           | ✔️       | -                 | 58.9$_{+0.8}$ | 66.1$_{+0.5}$ |
> | Ours (CenterPoint) | ❌       | -   | 61.5 | 68.0 |
> | Ours (CenterPoint) | ✔️       | pixel semantics   | 64.2$_{+2.7}$ | 69.7$_{+1.7}$ |
>
>
> Table 2:
>
> | Method              | PreTrain | Pixel Signal Used        | mAP      |
> |:-------------------:|:--------:|:------------------------:|:--------:|
> | PointRCNN           | ❌       | -                        | 28.74    |
> | SECOND              | ❌       | -                        | 51.89    |
> | CenterPoint         | ❌       | -                        | 60.05    |
> | PointContrast       | ❌       | -                        | 51.89    |
> | PointContrast       | ✔️       | -                        | 53.59$_{+1.70}$ |
> | SLidR               | ❌       | - | 28.80    |
> | SLidR               | ✔️       | super-pixel & pixel feature | 30.72$_{+1.92}$ |
> | ProposalContrast    | ❌       | -                        | 64.24    |
> | ProposalContrast    | ✔️       | -                        | 66.32$_{+2.08}$ |
> | GD-MAE              | ❌       | -                        | 62.62    |
> | GD-MAE              | ✔️       | -                        | 64.92$_{+2.30}$ |
> | Ours (CenterPoint)  | ❌       | -           | 64.28    |
> | Ours (CenterPoint)  | ✔️       | pixel semantics           | 67.41$_{+3.13}$ |
>
> We hope these amendments will help readers better understand the prerequisites of our method and its comparison with others.
>
> ---
>
> **Q3: The generalizability of using 2D segmentation models.**
>
> **A3:** Our method leans on well-trained 2D segmentation models. However, given their widespread adoption spanning a diverse spectrum of applications, we believe this reliance is justified.
>
> Notably, with the recent advancements in Vision Foundation Models (VFMs), we envision leveraging these VFMs to further enhance our method's generalization capabilities. For instance, SAM [60], a current popular segmenter known for its strong generalization performance, could be integrated into our framework. While SAM's predictions exclude semantic labels, its semantic features present an attractive supervisory signal, marking an exciting trajectory for our ensuing endeavors.
>
> Moreover, our method has consistently demonstrated its efficacy across a range of datasets, benchmarks, and tasks, underscoring the inherent robustness ingrained within our approach. Nevertheless, we will try to explore the feasibility of utilizing VFMs to further enhance the generalization ability.
>
> [60] SAM: Segment Anything.
>
> ---
>
> **Q4: Implications of potential 2D segmenter failures.**
>
> **A4:** In instances where the 2D segmenter fails to accurately predict semantics, a potential impact on the quality of pre-training arises due to the methodology's reliance on 2D semantic labels. However, our extensive empirical evaluations have consistently demonstrated the segmenter's resilience across a diverse array of datasets. This robustness can be attributed to its foundation on the Cityscape dataset, which includes scenes similar to those present in other datasets.
>
> Nonetheless, we acknowledge there might be situations where the 2D segmenter could be unreliable. In such cases, our approach incorporates the maximum prediction score as a weighting factor within our loss function, as detailed in Lines 127-129 of our methodology section. This weighting scheme effectively assigns reduced significance to potentially erroneous semantic labels, thereby mitigating the potential consequences of segmentation errors on the overall pre-training process.

---

> > ### Comment · Reviewer_8P8s · 2023-08-20
> > **Keep my rating**
> >
> > Thanks to the reviewer for responding to my comments with further comparative studies and analyses. The practice of utilizing 2D output to aid in training 3D tasks is widely accepted. The remaining issue pertains more to the constraint of denoising and enhancing the generalizability of the 2D views, which are restricted. The authors have indeed taken into account my comments, yet the novelty of the paper remains somewhat constrained. I intend to maintain my current rating.

---

> > > ### Author Response · Authors · 2023-08-20
> > > **Thanks for your comments**
> > >
> > > Thank you for your feedback.  In this study, we introduce a semantic rendering approach for pre-training point clouds that effectively tackles the challenges associated with reconstruction ambiguities and occlusions. Across a diverse range of datasets and tasks, our method consistently demonstrates improvements over various baseline methods. We will further revise the paper accordingly.

---

### Official Review · Reviewer_4GhG · 2023-07-06

**Soundness:** 3 good
**Presentation:** 3 good
**Contribution:** 3 good
**Rating:** 8
**Confidence:** 5

**Summary:**

This work proposes a new pretraining algorithm for outdoor 3D perception tasks, where images are utilized to provide comprehensive semantic information. The main algorithm is to leverage the semantics of images for supervision through neural rendering. The authors also apply point-wise masking with a high mask ratio to further enhance performance. The pretraining brings notable performance gains on multiple benchmarks.

**Strengths:**

1.	The authors provide a novel insight into the exploitation of image semantics by combing off-the-shelf image segmenter and neural rendering, which I think is well-motivated.
2.	Extensive experiments on multiple benchmarks demonstrate the effectiveness of the proposed method.
3.	The paper is well-written and easy to follow. The illustrations are straightforward and helpful.

**Weaknesses:**

1. I acknowledge the motivation and the method design of this paper. Different from neural rendering for RGB as NeRF does, the high-level idea of rendering semantics from point-cloud is quite novel. However, I have heavy concerns about the use of Cityscape pre-trained segmenter as it is not general enough, e.g., Cityscape pre-trained segmenter can perform well on nuscenes and Once but can't on Waymo. Despite the authors explaining the reason Waymo experiments do not perform that well on Line101 of the appendix, I think the poor generalizability of pre-trained segmenter maybe also a reason. So I suggest replacing the pre-trained segmenter with SAM.

2. This work shares a similar spirit to Ponder [1]. Despite [1] for indoor scenes, the paper aims at outdoor scenes, the authors should carefully discuss the intrinsic differences.

[1] Ponder: point-cloud pre-training via neural rendering.

**Questions:**

Please refer to the weakness part, I would raise my score if the two concerns can be addressed.

**Limitations:**

Please refer above.

---

> ### Author Rebuttal · Authors · 2023-08-09
>
> We appreciate your positive feedback regarding the motivation and results of our research. We understand your concerns about certain aspects of our paper and would like to provide some clarification.
>
> **Q1: On the selection of the pre-trained segmenter.**
>
> **A1:** We will definitely explore the possibility of replacing the segmenter with SAM. The selection of the pre-trained segmenter on Cityscape was motivated by its contextual relevance to our experimental datasets—all involve autonomous driving scenarios. The diverse array of scenes within Cityscape aligns with those found in other datasets, thus enhancing the segmenter's ability to generalize across a range of autonomous driving datasets.
>
> In the case of Waymo, its point cloud exhibits greater density compared to other datasets, attributed to the utilization of 64-beam LiDAR scanning, whereas datasets like nuScenes employ 32-beam LiDARs. Moreover, Waymo only includes images captured from front and side views, whereas images in other datasets encompass a wider range of perspectives. As a result, the utility of these images in enhancing point cloud downstream tasks, such as object detection, remains limited on the Waymo dataset.  This observation also provides an explanation for the relatively modest improvements achieved through image-derived semantic information during the pre-training phase.
>
> Nonetheless, we agree with you that SAM might exhibit superior generalization performance compared to the segmenter trained on Cityscape. While SAM exclusively predicts object masks without providing semantic labels, an intriguing strategy could involve substituting semantic rendering with semantic feature rendering. In this scenario, SAM could be employed to extract semantic features from images to serve as the supervision. This endeavor indeed holds great promise, and we deeply value your insightful suggestion. In future work, we will consider employing Vision Foundation Models like SAM to enhance the generalizability of our pre-training framework.
>
> ---
>
> **Q2: The intrinsic differences between our work and Ponder.**
>
> **A2:** Several fundamental differences indeed exist between our work and Ponder, and it's crucial to emphasize these.
>
> - First and foremost, the application domains differ: Ponder focuses on indoor scenes, where point clouds often contain color information, facilitating color-based pre-training supervision. In contrast, our work addresses outdoor environments—specifically, autonomous driving—where point clouds are typically LiDAR-derived and colorless. Consequently, Ponder is not applicable due to the absence of color data. In this context, we propose semantic rendering. Unlike Ponder's color-based rendering, our approach capitalizes on the semantic consistency between point clouds and images, offering a distinct strategy for point cloud pre-training.
> - Second, the pixel sampling strategies differ: Unlike the one-to-one point cloud-to-pixel correspondence found in densely scanned indoor point clouds, outdoor point clouds exhibit only partial semantic information from the image due to their sparsity. Furthermore, the semantic imbalance prevails within the point clouds. Our method addresses these challenges by sampling pixels projected from point clouds with a class-balanced strategy. This stands in contrast to Ponder's approach of random sampling from images.
> - Lastly, Ponder employs a 3D voxelized feature volume for rendering, whereas we favor a bird's eye view (BEV) feature-based rendering strategy better suited to the unique traits of outdoor settings.
>
> We will discuss Ponder in our revision.

---

> > ### Comment · Reviewer_4GhG · 2023-08-16
> >
> > Thanks for the rebuttal. I would prefer to provide the results around SAM. Although SAM does not provide semantic labels, could you use "Semantic-Segment-Anything", "Recognize Anything", and/or "SAM+OV-seg (https://huggingface.co/spaces/facebook/ov-seg)". If the model can be adapted to these models and it is demonstrated in the rebuttal phase, it will be very strong to accept.
> >
> > My concerns about differences to Ponder are addressed. Thanks!

---

> > > ### Author Response · Authors · 2023-08-20
> > > **Adaptation to SAM-Based Segmenter and Performance Prospects**
> > >
> > > Thanks for your constructive suggestions. We have implemented the method you suggested, substituting the segmenter with a SAM-based approach. Given the ease of use and time constraints, we opted to employ Semantic-Segment-Anything (SSA) as the segmenter. The results are outlined below. Apart from the segmenter, all other experimental parameters remain consistent with those detailed in Section 4.4.
> > >
> > > | Segmenter | PreTrain | mAP | NDS |
> > > |:----------------:|:--------:|:-----------------:|:--------:|
> > > | Baseline | ❌     | 61.5 | 68.0 |
> > > | DeepLabv3 | ✔️       | 64.2$_{+2.7}$ | 69.7$_{+1.7}$ |
> > > | Semantic-Segment-Anything | ✔️     | 64.5$_{+3.0}$ | 69.9$_{+1.9}$   |
> > >
> > > Thanks to SAM's strong generalization capabilities and segmentation performance, our method demonstrated further enhancements when paired with SSA as the segmenter. We anticipate that fine-tuning hyperparameters will lead to even more substantial performance improvements.

---

> > > > ### Comment · Reviewer_4GhG · 2023-08-20
> > > > **Raise my score to strong accept**
> > > >
> > > > Cool! Previously I would suggest the work is restricted to pre-trained segmentor. However given the significantly development of vision foundation model, the method can seamlessly make use of them. Please do put these experiments in the later version. I like it and will raise my score to strong accept. Please also make sure putting more related works that introducing volumetric rendering into perception tasks, such as previously mentioned Ponder, as well as recent NeRF-Det [1] and NeSF [2] etc.
> > > >
> > > > [1] NeRF-Det: Learning Geometry-Aware Volumetric Representation for Multi-View 3D Object Detection. ICCV 2023.
> > > > [2] NeSF: Neural Semantic Fields for Generalizable Semantic Segmentation of 3D Scenes. TMLR 2022.

---

> > > > > ### Author Response · Authors · 2023-08-21
> > > > > **Thanks for your positive feedback!**
> > > > >
> > > > > Thank you very much for the positive feedback!
> > > > >
> > > > > We will incorporate these experiments and related works into the revised version. Your insightful comments and suggestions greatly contribute to enhancing the quality of our paper. Thanks!

---

### Official Review · Reviewer_4EsX · 2023-07-08

**Soundness:** 2 fair
**Presentation:** 3 good
**Contribution:** 3 good
**Rating:** 5
**Confidence:** 3

**Summary:**

1. The paper proposes PRED, a novel pre-training framework for outdoor point clouds that leverages image semantics through neural rendering. The paper addresses the challenges of incompleteness and occlusion in point clouds, which are common in outdoor LiDAR datasets for autonomous driving.
2. The paper uses an encoder-decoder architecture to extract a BEV feature map from the point cloud and render semantic maps from image views, supervised by image segmentation and depth estimation.
3. The paper incorporates point-wise masking with a high mask ratio (95%) to enhance the pre-training performance and avoid losing semantics of small objects.
4. The paper conducts extensive experiments on nuScenes and ONCE datasets, showing that PRED significantly improves various baselines and state-of-the-art methods on 3D object detection and BEV map segmentation tasks.

**Strengths:**

1. The paper presents a novel pre-training framework for outdoor point clouds that integrates image semantics through neural rendering. This is a creative and effective way to address the incompleteness and occlusion issues in point clouds, which are often overlooked by previous pre-training methods. The paper also introduces point-wise masking with a high mask ratio, which is different from the conventional patch-wise or group-wise masking strategies and preserves more semantics of small objects.
2. The paper is technically sound and well-motivated. The paper provides a clear and detailed description of the proposed method, including the encoder-decoder architecture, the semantic rendering pipeline, the loss functions, and the masking strategy. The paper also conducts extensive experiments on two large-scale datasets, nuScenes and ONCE, and compares with various baselines and state-of-the-art methods on 3D object detection and BEV map segmentation tasks. The paper reports significant improvements over previous methods, demonstrating the effectiveness and generality of the proposed framework.

**Weaknesses:**

1. The paper does not conduct ablation studies on the choice of image segmentation model and its impact on the pre-training performance. The paper uses DeepLabv3 as the image segmenter but does not justify or evaluate this choice. It is unclear how the quality and accuracy of the image segmentation model affect the semantic rendering and supervision.
2. This paper does not report the computational cost or time complexity of the pre-training framework. Semantic rendering involves sampling and aggregating points along multiple rays for each pixel, which may be computationally expensive and memory-intensive. It would be helpful to provide some statistics or benchmarks on the pre-training speed and resource consumption.

**Questions:**

Please see the comments on 'weaknesses'.

**Limitations:**

Please see the comments on 'weaknesses'.

---

> ### Author Rebuttal · Authors · 2023-08-09
>
> We're gratified by your acknowledgment of our approach's novelty and efficacy. We also value the concerns you've raised and here's our detailed response:
>
> **Q1: Evaluating the Impact of Image Segmentation Model Choices on Pre-training Performance.**
>
> **A1:** We acknowledge the importance of examining the choice of image segmentation model. To this end, we conducted ablation studies involving a range of renowned segmentation models, namely PSPNet [58], DeepLabv3 (MobileNet), DeepLabv3 (ResNet101), and SegFormer [59]. Keeping other settings consistent with Section 4.4, our results (as presented below) demonstrated robustness in pre-training performance across these models. DeepLabv3 ultimately emerged as our preferred choice, owing to its impressive performance and user-friendliness. These results will be incorporated into our revision.
>
>
> | Image Segmenter           | PreTrain | mAP      | NDS      |
> |:-------------------------:|:--------:|:--------:|:--------:|
> | baseline                  | ❌       | 61.5     | 68.0     |
> | PSPNet                    | ✔️       | 63.9$_{+2.4}$ | 69.5$_{+1.5}$ |
> | DeepLabv3 (MobileNet)     | ✔️       | 64.2$_{+2.7}$ | 69.7$_{+1.7}$ |
> | DeepLabv3 (ResNet101)     | ✔️       | 64.4$_{+2.9}$ | 69.7$_{+1.7}$ |
> | SegFormer                 | ✔️       | 64.3$_{+2.8}$ | 69.9$_{+1.9}$ |
>
> [58] PSPNet: Pyramid Scene Parsing Network.\
> [59] SegFormer: Simple and Efficient Design for Semantic Segmentation with Transformers.
>
> ---
>
> **Q2: Computational cost or time complexity of the pre-training framework.**
>
> **A2:** Sorry for overlooking these details earlier, we will incorporate this information into the revised version. For our pre-training scheme, we utilized eight V100 GPUs, each equipped with 32GB memory. Abiding by the configurations detailed in Section 4.1 and Appendix A, and capitalizing on mixed precision training, each GPU consumed approximately 28GB to 30GB of memory. A complete pre-training exercise spanning 45 epochs on the nuScenes dataset was accomplished in around 32 hours. Additionally, training across the ONCE datasets—20 epochs for the small variant, 5 for the medium, and 3 for the large—required approximately 30, 38, and 45 hours, respectively. Since we only sample 768 pixels per scene as a training batch in each iteration, the overall training time and resource consumption are acceptable.

---

> > ### Comment · Reviewer_4EsX · 2023-08-22
> >
> > I would thank the authors for their rebuttal. I will retain my positive score.

---

### Decision · Program_Chairs · 2023-09-21

**Decision:**

Accept (poster)

**Comment:**

The paper proposes PRED, a novel pre-training framework for outdoor point clouds that leverages image semantics through neural rendering. All reviewers reach concensus that the paper should be accepted. AC read the paper, rebuttal, author response and discussions. The paper well addressed reviewer concerns and additionaly provide more experimental results as requested.
Please revise the paper accordingly based reviewer comments for the camera-ready.